# A glycosylated lipooctapeptide promotes uptake and growth of *Mycobacterium abscessus* in the host

Louis David Leclercq [1,2], Vincent Le Moigne [2,11], Wassim Daher [3,4,11], Mélanie Cortes [2], Bertus Viljoen [3,10], Yara Tasrini [3], Xavier Trivelli [5], Hélène Lavanant [6], Isabelle Schmitz-Afonso [6], Nicolas Durand [2], Franck Biet [7], Yann Guérardel [1,8] ✉, Laurent Kremer [3,4] ✉ & Jean-Louis Herrmann [2,9] ✉

Pathogenic mycobacteria produce a wide array of lipids which participate in host cell interactions and virulence. While some of these are conserved across all mycobacteria, others, like glycopeptidolipids (GPL), are restricted to a few species. *Mycobacterium abscessus*, an emerging rapid-growing pathogen, transitions from a smooth to a virulent rough variant upon the loss of surface GPL. Here, we discovered that *M. abscessus* and phylogenetically-close species harbor a second GPL-related locus, comprising two adjacent non-ribosomal peptide synthetase genes, *MAB_4690c* and *MAB_4691c*. A *MAB_4690c* deletion mutant (Δ*MAB_4690c*) failed to produce a yet undescribed lipid, designated GL8P for glycosylated lipooctapeptide, sharing an acylated octapeptide core adorned by mono or di-*O*-rhamnosyl substituents. Δ*MAB_4690c* exhibited impaired uptake and survival in THP-1 cells and was attenuated in mice. Importantly, GL8P elicited a strong humoral response in patients infected with *M. abscessus*. These results highlight the role of GL8P in the pathophysiology of infection by rough *M. abscessus* and suggest its potential as a selective marker for *M. abscessus* infections.

Pathogenic microorganisms have evolved mechanisms to undergo morphotype transitions to evade host immune responses[1,2]. This can occur through antigenic variation, as observed in *Borrelia*, *Neisseria* or *Trypanosoma*[1–3]. Alterations to the structure or composition of the outer membrane can also facilitate immune evasion and establish successful infection, as proposed for *Mycobacterium tuberculosis*[4] or as reported in cystic fibrosis (CF)-related lung infections by mucoid *Pseudomonas*

*aeruginosa* isolates and rough (R) variants of *Mycobacterium abscessus*[5–9]. These morphotype transitions may also confer increased resistance to antimicrobial agents and enhanced pathogenicity.

*M. abscessus*, an opportunistic and rapidly-growing non-tuberculous mycobacterium (NTM), is currently under intense investigation owing to its extreme resistance to most antibiotic classes[10,11]. It causes nosocomial or health-associated cutaneous infections[12] and severe

[1]Université Lille, CNRS, UMR 8576—UGSF—Unité de Glycobiologie Structurale et Fonctionnelle, Lille, France. [2]Université Paris-Saclay, UVSQ, Inserm, Infection et inflammation, Montigny-Le-Bretonneux, France. [3]Centre National de la Recherche Scientifique UMR9004, Institut de Recherche en Infectiologie de Montpellier (IRIM), Université de Montpellier, Montpellier, France. [4]INSERM, IRIM, Montpellier, France. [5]Université de Lille, CNRS, INRAE, Centrale Lille, Université d'Artois, FR 2638—IMEC—Institut Michel-Eugène Chevreul, Lille, France. [6]Normandie Univ, Univ Rouen Normandie, CNRS, CARMeN UMR 6064 (ex-COBRA-LCMT), Rouen, France. [7]INRAE, UMR ISP 1282, Université de Tours, Nouzilly, France. [8]Institute for Glyco-core Research (iGCORE), Gifu University, Gifu, Japan. [9]AP-HP, Service de Microbiologie, GHU Paris Saclay, Hôpital Raymond Poincaré, Garches, France. [10]Present address: IPBS, CNRS, Toulouse, France. [11]These authors contributed equally: Vincent Le Moigne, Wassim Daher. ✉e-mail: yann.guerardel@univ-lille.fr; laurent.kremer@irim.cnrs.fr; jean-louis.herrmann@aphp.fr

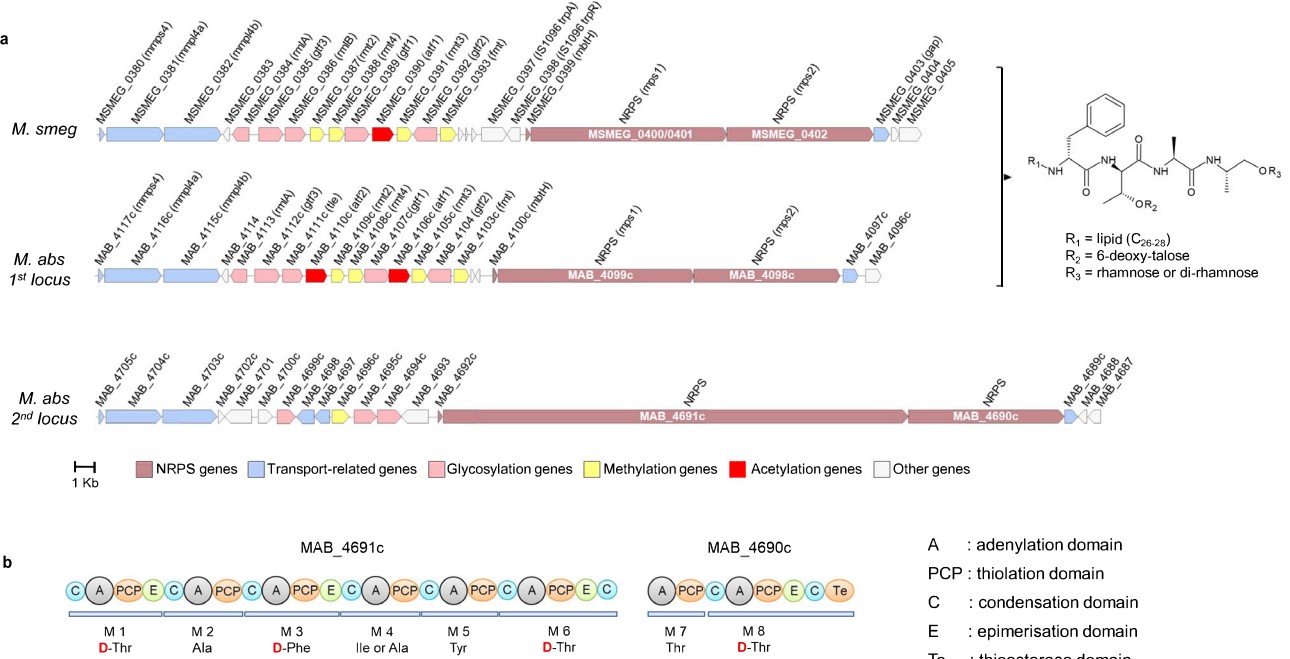

**Fig. 1 | In silico analysis of a second GPL locus and prediction of the amino acid sequence produced by adjacent NRPS. a** Genomic organization of the GPL locus in *M. smegmatis* and *M. abscessus* (*M. abs* 1st locus) and of a new GPL-like biosynthetic gene cluster that includes the *MAB_4690c* and *MAB_4691c* NRPS genes with transporter (*mmpL*) and glycosyltransferase genes, and regulators (*M. abs* 2nd locus). **b** Prediction of the amino acids added by the NRPS MAB_4690c and MAB_4691c. Each module contains an adenylation domain (A) to activate the amino acid, a thiolation domain (PCP), which is esterified by the activated amino acid, and a condensation domain (C) that transfers the amino acid to elongate the peptidic backbone. Four non-mandatory epimerases domains (E) and the thioesterase domain (Te) that releases the free peptide are annotated.

pulmonary diseases, particularly afflicting patients with CF or bronchiectasis[13,14], with emerging circulating clones contributing to epidemics[12,15,16]. *M. abscessus* is the first NTM for which human-to-human transmission has been proposed[16,17]. During infection, *M. abscessus* shifts from a smooth (S), colonizing, biofilm-forming variant to a rough (R) non-biofilm-forming, non-motile but cord-forming variant[7,18]. The S and R variants represent two forms of the same isolate, which can co-exist in the patient and evolve differently in response to host immune pressure. The S-to-R transition is associated with a more virulent phenotype[6,19–21] causing severe and acute diseases, characterized by a robust pro-inflammatory response[6–8,18]. Mechanistically, this transition is associated with the depletion of major surface glycopeptidolipids (GPL)[22–25], produced by several NTMs including *Mycobacterium smegmatis*, *Mycobacterium chelonae*, *M. abscessus* and slow-growing species like *Mycobacterium avium* subsp. *avium*[26]. GPL are major surface-exposed lipids[26,27] playing a crucial role in sliding motility, biofilm formation, cell wall integrity[24,28,29] and pathogenicity[24,30]. Genomic and transcriptomic studies have unraveled DNA insertions/deletions in the R forms, mainly in genes encoding non-ribosomal peptide synthases (NRPS) and membrane proteins involved in GPL transport[24,31–33]. Removal of GPL leads to highly aggregative and hydrophobic bacteria[24,30], associated with enhanced expression and surface localization of lipoproteins[34].

Here, in silico analysis identified a large locus containing genes encoding NRPS (*MAB_4690c* and *MAB_4691c*), potentially involved in the synthesis of a previously unreported GPL-related component. Genetic and structural analyses elucidated the structure of this lipid family, consisting of glycosylated lipooctapeptides (GL8P). Deletion of *MAB_4690c* in *M. abscessus* R, resulting in the loss of GL8P, caused strong attenuation in both cellular and animal models. Additionally, GL8P was detected in the sera of CF patients infected with *M. abscessus*, confirming its synthesis during host infection. Overall, this work provides insights into the structural lipid variability in mycobacteria

and highlights the important role of a lipid family in infection by these bacteria.

## Results

### In silico analysis of the *M. abscessus* genome identified a second GPL locus

Genomic analyses employing machine-learning methods in bioinformatics enable the identification of biosynthetic gene clusters encoding non-ribosomal peptide synthase (NRPS) enzymes responsible for metabolite core synthesis and regulatory transporters essential for GPL metabolite production. Together with antiSMASH 7.0, these tools uncovered multiple biosynthetic gene clusters loci from the complete genomic sequence of *M. abscessus* subsp. *abscessus* ATCC 19977 strain. Further analysis confirmed the presence of two biosynthetic gene clusters separated by 600 kbp, responsible for the production of two structurally different GPLs. In addition to the classical GPL locus (*MAB_4097c* to *MAB_4117c*)[22], a second locus (*MAB_4689c* to *MAB_4705c*), encompasses two tandem genes, *MAB_4690c* and *MAB_4691c*, encoding bimodular and hexamodular NRPS (Fig. 1a). Bioinformatic predictions suggest that they are responsible for the synthesis of an octapeptide that constitutes the core of a yet unreported GPL-like component, hereafter referred to as GL8P (Fig. 1b). Furthermore, the prediction of epimerization domains and the presence of four epimerases strongly suggest that amino acids at positions 1, 3, 6, and 8 are in D configuration. Additionally, bioinformatic tools predicted the production of a possible *N*-acylated octapeptide with a core sequence corresponding to D-Thr-L-Ala-D-Phe-(L-Ile/L-Ala)-L-Tyr-D-Thr-L-Thr-D-Thr.

Surrounding the NRPS genes are several genes likely participating in the synthesis, transport, and regulation of GL8P, encoding glycosyltransferases, methyltransferases, an MbtH-type protein, MmpL and MmpS membrane proteins, lipases, a fatty-acid CoA ligase, glyoxalases, and several TetR-type transcriptional regulators (Fig. 1a).

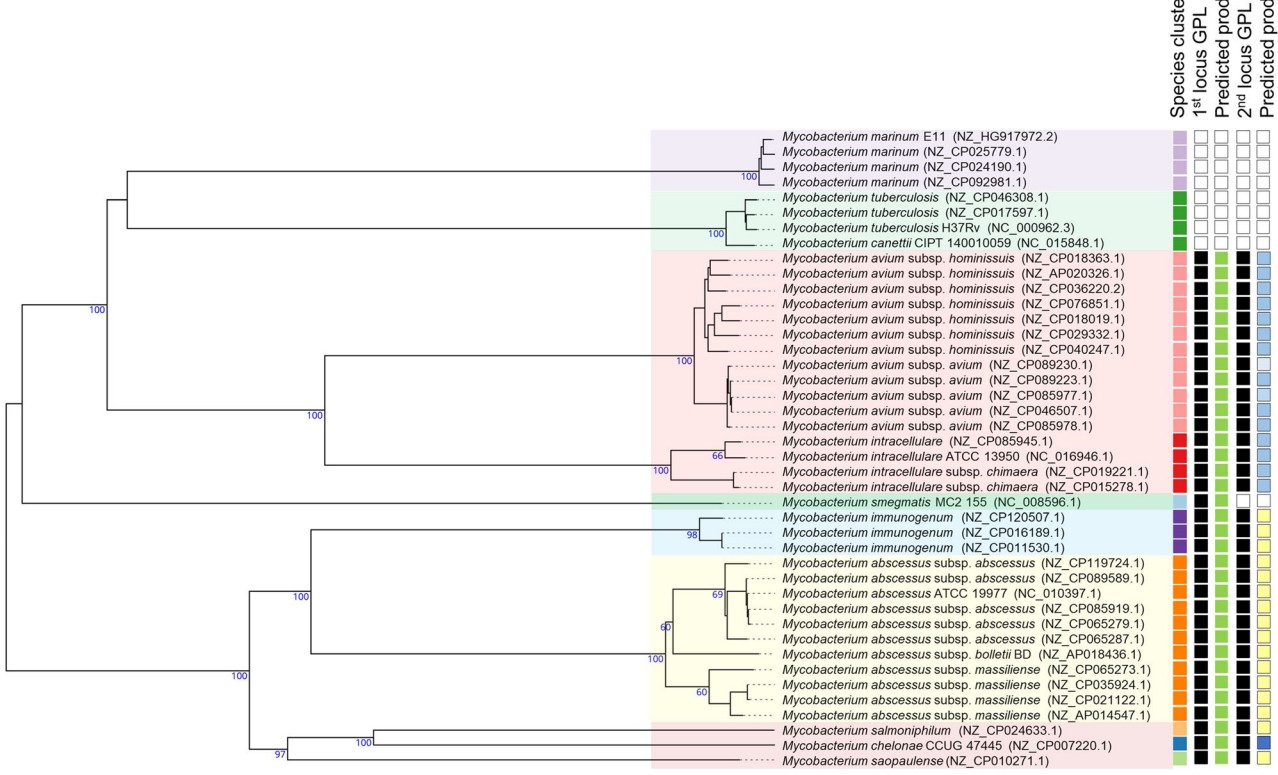

**Fig. 2 | Occurrences of the two GPL loci in mycobacterial genomes.** The phylogenetic tree was inferred from GBDP distances calculated from complete genomic sequences of 42 mycobacterial strains. For the 1st GPL locus, bioinformatic analysis of NRPSs shows that the deduced product is ubiquitous and identical in all GPL-producing mycobacteria (green squares). However, not all GPL-producing mycobacteria systematically have a second locus. For this one, the nature of the deduced product differs, as illustrated by the different colored squares. Note that this second locus is conserved in the *M. abscessus* complex and in only closely related phylogenetic strains (yellow squares). Color code used: For the "Species column", each species is differentiated by a different color. For the "locus" columns, white for absence or black for presence. For the "Product" columns, each color represents a different predicted structure (Data S1).

AntiSMASH was used to identify and compare NRPS's and their products inferred from complete mycobacterial genomes. Examination of "GPL-like" biosynthetic gene clusters revealed that in mycobacterial species originally described as GPL producers, the standard "GPL-1" locus was conserved, while one additional locus was found only in some of them. However, the composition and size of the NRPS in this second locus and their predicted products vary across species (Data S1). This indicates that the GPL synthesized by the second locus is unique and conserved only in the *M. abscessus* complex (MABSC) and in a small number of phylogenetically-related species, such as *Mycobacterium immunogenum* (Fig. 2).

### *MAB_4690c* is essential for the production of two novel glycopeptidolipids

To assess the role of MAB_4690c in lipid biosynthesis, *MAB_4690c* was disrupted by introducing a zeocin resistance cassette in *M. abscessus* R using the recombineering strategy[35], resulting in R∆*4690c* (Table S1). Complementation was achieved by introducing a copy of the gene under the control of the *hsp60* promoter, yielding R∆*4690c::4690c*. Apolar (Fig. 3a) and polar (Fig. 3b, c) lipids were extracted from the parental R, R∆*4690c* and R∆*4690c::4690c* strains grown on Middlebrook 7H10 agar and analyzed by thin layer chromatography (TLC). The mono-*O*-acylated phosphatidylinositol dimannoside (Ac$_1$PIM$_2$), di-*O*-acylated phosphatidylinositol dimannoside (Ac$_2$PIM$_2$) and trehalose dimycolate (TDM) profiles were comparable in all three strains (Fig. 3a, b) while a glycolipid at a retardation factor (Rf) of 0.4 appeared more clearly in the polar lipid fraction of R∆*4690c::4690c* (Fig. 3c). Common phospholipids (PLs)

were also detected by TLC, but no quantitative differences were observed between the strains. In contrast, comparative matrix-assisted laser desorption/ionization-time of flight (MALDI-TOF) mass spectrometry (MS) analysis of the polar lipids revealed two intense signals at *m/z* 1392 and 1750 in R and R∆*4690c::4690c* but not in R∆*4690c* (Fig. 3d), whereas Ac$_1$PIM$_2$ at *m/z* 1446 (Fig. S1a–c) is observed in all three strains. The presence of MS signals at *m/z* 1392 and 1750 in R and R∆*4690c::4690c* was confirmed by liquid chromatography coupled to mass spectrometry (LC-MS) analyses that show intense extraction signals at time retentions (Tr) of 24.1 and 27.9 min (Fig. 3e, f). Similar MS data were obtained when growing the strains on LB agar (Fig. S1d–f). These results indicate that MAB_4690c is involved in the biosynthesis of two related lipids, designated GL8Pa and GL8Pb.

### GL8Pa and GL8Pb share a common octapeptide core integrating a *para-O*-methyl tyrosine

GL8Pa and GL8Pb were purified following multiple chromatographic steps. Both compounds were first separated from the co-extracted 3,4 di-*O*-acyl trehalose using silica gel flash chromatography, and their purity was determined through nuclear magnetic resonance (NMR) spectroscopy, MALDI-TOF, and gas chromatography-mass spectrometry (GC-MS) analyses (Fig. S2a–d). Subsequently, GL8Pa and GL8Pb were separated by reversed-phase flash chromatography, and a sufficient amount of GL8Pb was purified to allow NMR analysis (Fig. S2e, f). GL8Pb was first dissolved in CDCl$_3$/CD$_3$OD (2:1) and analyzed by $^1$H/$^1$H COSY, $^1$H/$^1$H NOESY, $^1$H/$^{13}$C HSQC, and $^1$H/$^{13}$C HMBC very high-field NMR experiments (Figs. S3a, b and S4a–h). The complete spin systems

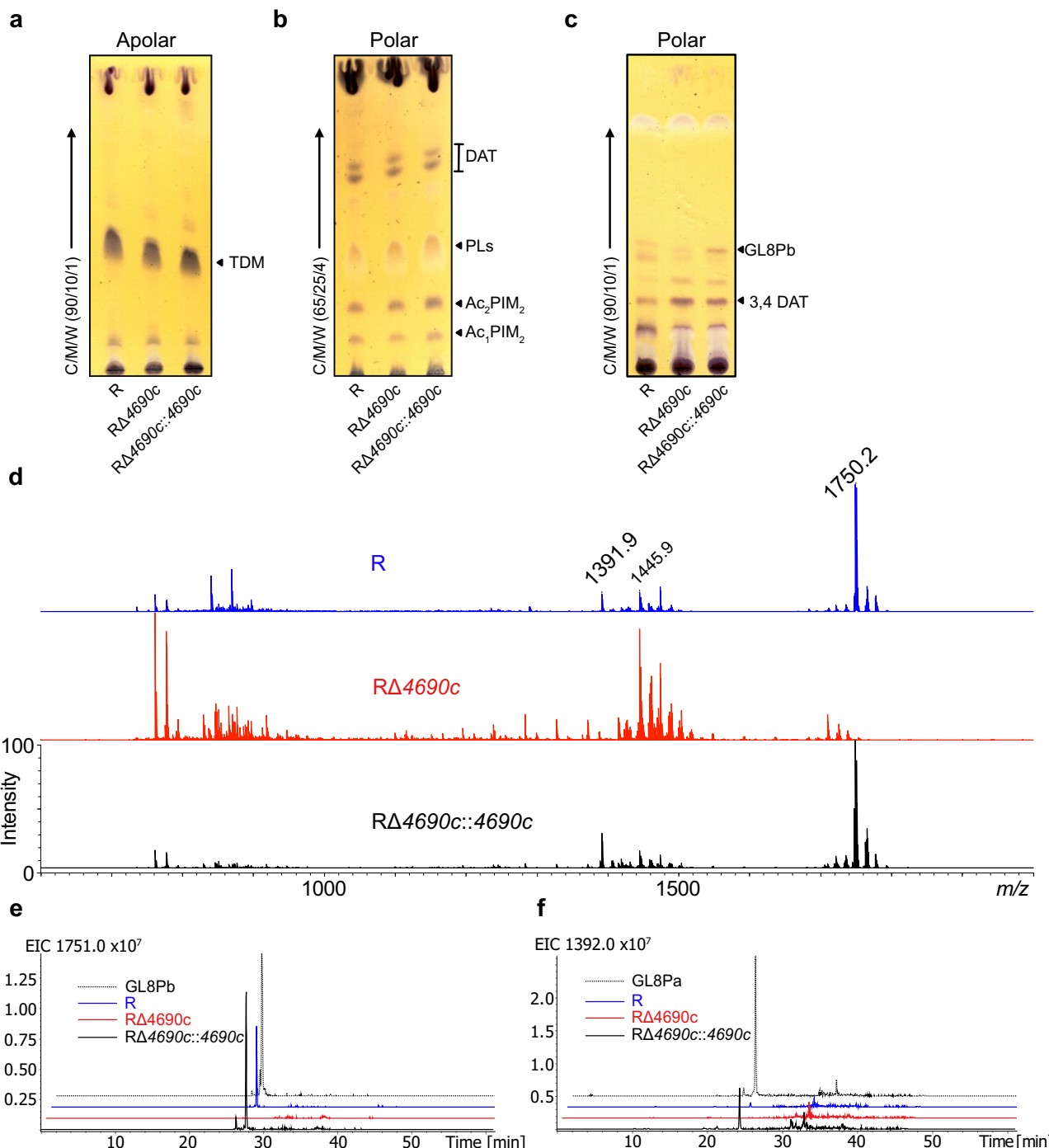

**Fig. 3 | Deletion of *MAB_4690c* results in the loss of two putative polar glyco-peptidolipids.** Apolar (**a**) and polar (**b**, **c**) lipids were extracted from bacteria grown on 7H10^OADC 7H10-OADC and separated by TLC using CHCl$_3$/CH$_3$OH/H$_2$O (C/M/W) solvent systems as indicated on the left. Identified lipids are indicated on the right side of each TLC. TDM Trehalose Dimycolate, Ac$_1$PIM$_2$ mono-*O*-acylated phosphatidylinositol dimannoside, PLs phospholipids, Ac$_2$PIM$_2$ di-*O*-acylated phosphatidylinositol dimannoside, DAT Di-*O*-Acyl Trehalose, 3,4 DAT 3,4 Di-*O*-Acyl Trehalose. 3,4 DAT was characterized in Fig. S2; GL8Pb, Di-*O*-rhamnosylated lipooctapeptide was characterized in Fig. 4. **d** MALDI-MS spectra in positive mode

of polar lipids from R (blue) and R$\Delta$*MAB_4690c::4690c* (black) show two signals at *m/z* 1392 (GL8Pa) and 1750 (GL8Pb), absent in R$\Delta$*4690c* (red). Conversely, ion at *m/z* 1446 corresponding to [M−H + 2Na]$^+$ adduct of Ac$_1$PIM$_2$ was detected in all strains. Its monosaccharide composition was confirmed by alditol acetate analysis (mannose, 2-*O*-methyl mannose, and *myo*-inositol)[68], and fatty acids methyl ester content (C$_{14}$/C$_{16}$/C$_{19}$) verified by GC-MS (data not shown) and MALDI-MS experiments (Fig. S1). Extracted ion chromatograms at *m/z* 1392 (**e**) and 1750 (**f**) of polar lipids show that GL8Pa and GL8Pb (dashed lines) are not detected in R$\Delta$*4690c* by LC-MS (Source data https://doi.org/10.5281/zenodo.14918824).

for the peptide and glycan moieties, reported in Table S2, allowed the identification of eight amino acids: three Thr, one Ala, one Phe, one *para-O*-methyl Tyr, one Ile, and one Leu. While characterizing the amino acid side chains, the tyrosine's substitution of Cζ with a methoxy group at 3.728/55.31 ppm was confirmed by $^1$H/$^1$H NOESY and

$^1$H/$^{13}$C HMBC experiments, revealing correlation from the CH$_3$O group to both the CHε position (Fig. S4g) and the Cζ position (Fig. S4h). The sequence of the octapeptide was established as T$_1$-A$_2$-F$_3$-I$_4$-Y$_5$-L$_6$-T$_7$-T$_8$, in general accordance with bioinformatic tools (except for L$_6$/T$_6$), and was further confirmed by $^1$H/$^1$H NOESY and TOCSY NMR experiments

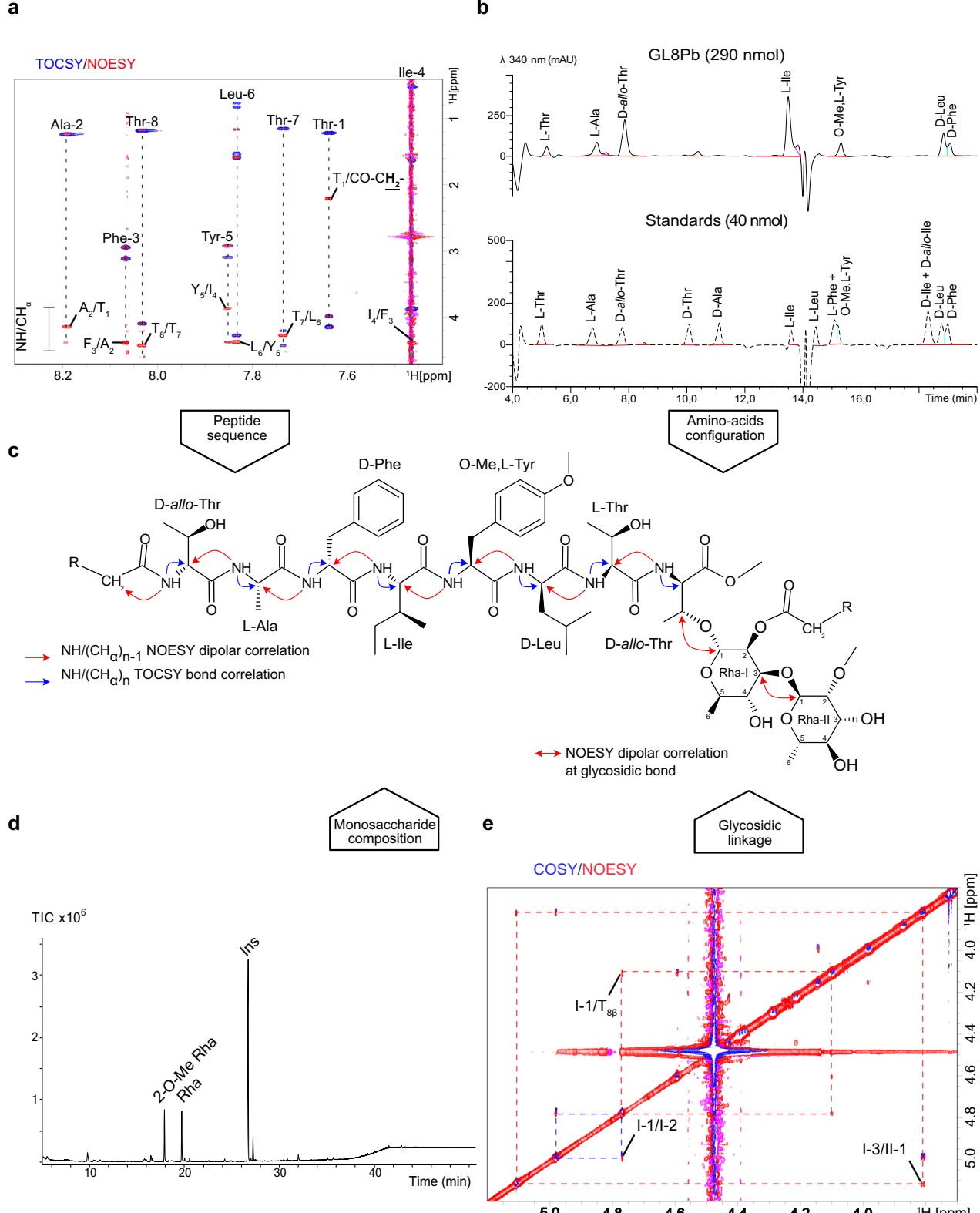

of partially deuterated glycolipid. This confirmation was facilitated by observing NOESY correlations between α protons of neighboring n-1 amino acids and COSY correlations between the α protons within each amino acid (Fig. 4a). The sequence was unambiguously confirmed by $^1H/^{13}C$ HMBC NMR experiment (Fig. S5a, b). To confirm the prediction of the D/L configuration of amino acids, we used Marfey's analysis, which involves derivatizing amino acids released from GL8Pb with a

reagent (here, 1-fluoro-2,4-dinitrophenyl-5-L-alanine amide) that reacts differently with D- and L-enantiomers. This allows them to be separated and identified using high-performance liquid chromatography (HPLC). We then compared the results with those of pure D and L amino acid standards. Our results indicate that *allo*-Thr, Phe, and Leu adopt a D conformation, while all other residues were L amino acids (Fig. 4b, c).

**Fig. 4 | Identification of the glycopeptide moiety of purified GL8Pb. a** The superposition of the $^1$H/$^1$H TOCSY (blue) and $^1$H/$^1$H NOESY (red) NMR spectra acquired in CDCl$_3$/CD$_3$OH (2:1) at 293 K shows the scalar (blue) and dipolar (red) correlations of amide protons with the α protons of amino acids at n and n-1 positions. Scalar correlations (blue) with β, γ, δ protons confirm the nature of each amino acid as well as the dipolar correlation between the amide proton of Thr-1 with the methylene from *N*-esterified acyl group. **b** Liquid chromatogram of Marfey's derivative free amino acids obtained after hydrolysis of purified GL8Pb. The derivatization of amino acids with a L-fluoro dinitrophenyl alaninamide (L-FDAA) produces either L- or D-amino acid/L-FDAA derivatives, which can be separated using a reverse phase HPLC column[69,70]. Comparison with amino acid standards shows that *allo*-threonine, leucine, and phenylalanine are D-amino acids. **c** Stereochemical structure of GL8Pb showing scalar (blue) and dipolar (red) correlations deduced from NMR analysis (A and E), as well as asymmetric carbon configuration. **d** GC-MS monosaccharide composition analysis of itol-acetates from purified GL8Pb shows two chromatographic signals at Rt 17.95 min and 19.78 min identified as 2-*O*-methyl rhamnose and rhamnose, respectively, based on their retention times and electronic impact mass spectra (Fig. S7). **e** Superimposed $^1$H/$^1$H COSY and $^1$H/$^1$H NOESY NMR spectra show cross-peaks between protons of Rha I-1 with β Thr-8 and of Rha II-1 with Rha I-3 (red dashed lines), confirming the nature of the two bonds (Source data https://doi.org/10.5281/zenodo.14918824).

NMR analysis also allowed us to establish the nature of the glycan moiety, starting from the two $^1$H/$^{13}$C anomer signals at 4.77/94.73 and 5.1/99.35 ppm on the $^1$H/$^{13}$C HSQC spectrum, assigned to two monosaccharides labeled I and II (Fig. S3a). The determination of the complete H1-H6 spin system and their associated $^3J_{H,H}$ and $^1J_{C,H}$ coupling constants provided useful clues to identify the nature of each monosaccharide. All information is summarized in Table S2. Residue I was identified as 2-*O*-acyl-α-rhamnopyranose, a finding confirmed by the deshielding of the I-2 proton at 4.98 ppm. In contrast, residue II was characterized as 2-*O*-methyl-α-rhamnopyranose, consistent with the deshielding of the II-2 carbon at 80.77 ppm. GC-MS analysis of itol-acetate derivatives further validated the presence of rhamnose and 2-*O*-methyl rhamnose (Figs. 4d and S5c). Then, the carbon deshielding of the CH$_β$ of Thr-8 at 72.12 ppm compared to Thr-1 and Thr-7 at 67.99 and 67.77 ppm suggests that the glycan moiety is linked to Thr-8 through an *O*-glycosidic bond. This was supported by the observation on the $^1$H/$^1$H NOESY NMR spectrum of a dipolar correlation between the I-1 proton at 4.77 ppm and the H$_β$ of Thr-8 at 4.1 ppm (Fig. 4e), as well as a strong $^3J_{H,C}$ correlations between those respective positions on the $^1$H/$^{13}$C HMBC spectrum (Fig. S5d). Similarly, strong NOESY and HMBC cross-peaks were observed between II-1 at δ 5.1/99.35 ppm and I-3 at δ 3.81/76.1 ppm, demonstrating that the 2-*O*-methyl-α-rhamnose residue II substitutes the 2-*O*-acylated-α-rhamnose residue I at the C3 position, consistent with the significant deshielding of I-3 carbon at 76.10 ppm compared to II-3 at 71.21 ppm (Figs. 4e and S5d). Together, these findings demonstrate that the 2-*O*-methyl-α-rhamnopyranosyl(1-3)2-*O*-acyl-α-rhamnopyranose disaccharide is connected to the β-hydroxyl group of Thr-8 (Fig. 4c).

To elucidate the nature of the acyl chains attached to Thr-1 and at the C2 position of the internal rhamnose residue I, GL8Pa and GL8Pb were further analyzed by mass spectrometry. Fourier transform ion cyclotron resonance (FT-ICR) MS analysis established the exact average monoisotopic mass of the GL8Pb sodium adduct at *m/z* 1750.065792 Da with a standard deviation of 2.27 10$^{-4}$ Da. Among the most probable raw formulas within an estimated maximum measurement error of 1 ppm, the crude formula $C_{90}H_{150}N_8O_{24}Na^+$ (estimated error of +0.16 ppm) aligns with the structure established by NMR (Figs. 5a and S6a). The comparison of the atomic composition with the previously determined di-*O*-rhamnosyl octapeptide suggested that the remaining lipid part contains 30 carbons, 58 hydrogens, and 3 oxygens. FT-ICR MS analysis showed that GL8Pa had a molecular mass of 1391,830005 Da, from which the $C_{71}H_{116}N_8O_{18}Na^+$ formula could be inferred with an estimated error of +0.02 ppm (Figs. 5b and S6b).

The MALDI QIT-TOF MS$^n$ experiments were used to (i) consolidate the amino acid sequence, (ii) confirm the nature and position of the carbohydrate moiety along the peptide, and (iii) infer the nature of the lipid moiety. For GL8Pb, the MS$^3$ fragmentation pattern of the parent ion at *m/z* 1750 > 1662 was dominated by N-terminal fragments a$_2$ to a$_6$, as well as C-terminal fragments y$_4$ to y$_7$, whose mass increments are in perfect agreement with the peptide sequence and the presence of an *N*-stearyl end with 18 carbons, 35 hydrogens, and 1 oxygen. At the C-terminal, the product ion at *m/z* 1140 suggests that the 2-*O*-methyl-α-rhamnopyranosyl(1-3)2-*O*-acyl-α-rhamnopyranose glycan motif is substituted by hydroxylauric acid, containing 12 carbons, 23 hydrogens, and 2 oxygens (Fig. 5a). Similar analysis of the compound at *m/z* 1778 revealed a lipoform differing from the compound at *m/z* 1750 by the presence of an *N*-arachidyl end instead of an *N*-stearyl group. (Fig. S6c). The MS$^3$ analysis of the GL8Pa parent ion at *m/z* 1391 > 1304 showed identical a and y fragmentation patterns as those for GL8Pb parent ion at *m/z* 1750 > 1662, demonstrating that both share a common peptide core and *N*-stearyl group (Fig. 5b). However, the 146 and 164 a.m.u decrease from fragment ions at *m/z* 1158 and 1140 suggests that the carbohydrate moiety of GL8Pa contains a non-acylated rhamnose. This structural difference between GL8Pa and GL8Pb aligns with the difference (-19 carbons, -34 hydrogens, -6 oxygens) observed in their crude formulas determined by FT-ICR.

## R∆*4690c* exhibits reduced internalization and survival in THP-1 cells and zebrafish

Infection studies using the human THP-1 cells were conducted to examine the intracellular behavior of the *M. abscessus* R, R∆*4690c*, and R∆*4690c::4690c* strains over time. Colony forming unit (CFU) enumeration revealed a marked reduction in bacterial load for R∆*4690c* relative to R and R∆*4690c::4690c*, beginning as early as 4 hpi and persisting at later time points (24 and 72 hpi), indicating a reduced intracellular burden of the mutant strain (Fig. 6a). This intracellular growth defect was corroborated by fluorescence microscopy, which quantified the proportion of infected cells. The percentage of R∆*4690c*-infected THP-1 cells was approximately half that of the R and R∆*4690c::4690c* strains at 4, 24, and 72 hpi (Fig. 6b). In addition, at 24 hpi, quantitative raw integrated density measurements displayed no statistically significant difference between R and R∆*4690c*. However, by 72 hpi, fluorescence intensity for R∆*4690c* decreased by ~50%, indicating a reduced bacterial density within THP-1 cells and highlighting an impairment in the long-term growth of the mutant strain (Fig. 6c). Fluorescence imaging further revealed an early defect in bacterial uptake by THP-1 cells at 4 hpi, as R∆*4690c* internalization was reduced compared to R and R∆*4690c::4690c* (Fig. 6d). At 72 hpi, fluorescence microscopy highlighted a pronounced growth defect for R∆*4690c*, further emphasizing the inability of the mutant to maintain intracellular proliferation (Fig. 6e). Next, to assess the role of actin dynamics in bacterial adhesion and internalization, cells were treated with Cytochalasin D. While Cytochalasin D treatment failed to affect the adhesion of the three strains to the cells which were kept on ice (Fig. 6f), it substantially inhibited their internalization when cells were incubated at 37 °C (Fig. 6g), confirming the importance of actin polymerization in bacterial uptake. Collectively, these results support the implication of GL8P, either directly or indirectly, in adhesion of *M. abscessus* to the cell surface.

Parallel studies using zebrafish embryos have provided additional insights into the host-pathogen dynamics of *M. abscessus*, specifically exploring interactions with the host innate immune system. These models have highlighted the critical role of macrophages in controlling infection, emphasizing the importance of innate immunity in managing bacterial growth and persistence[20,36]. Embryos displayed a significantly lower bacterial burden when infected with R∆*4690c*

**a**

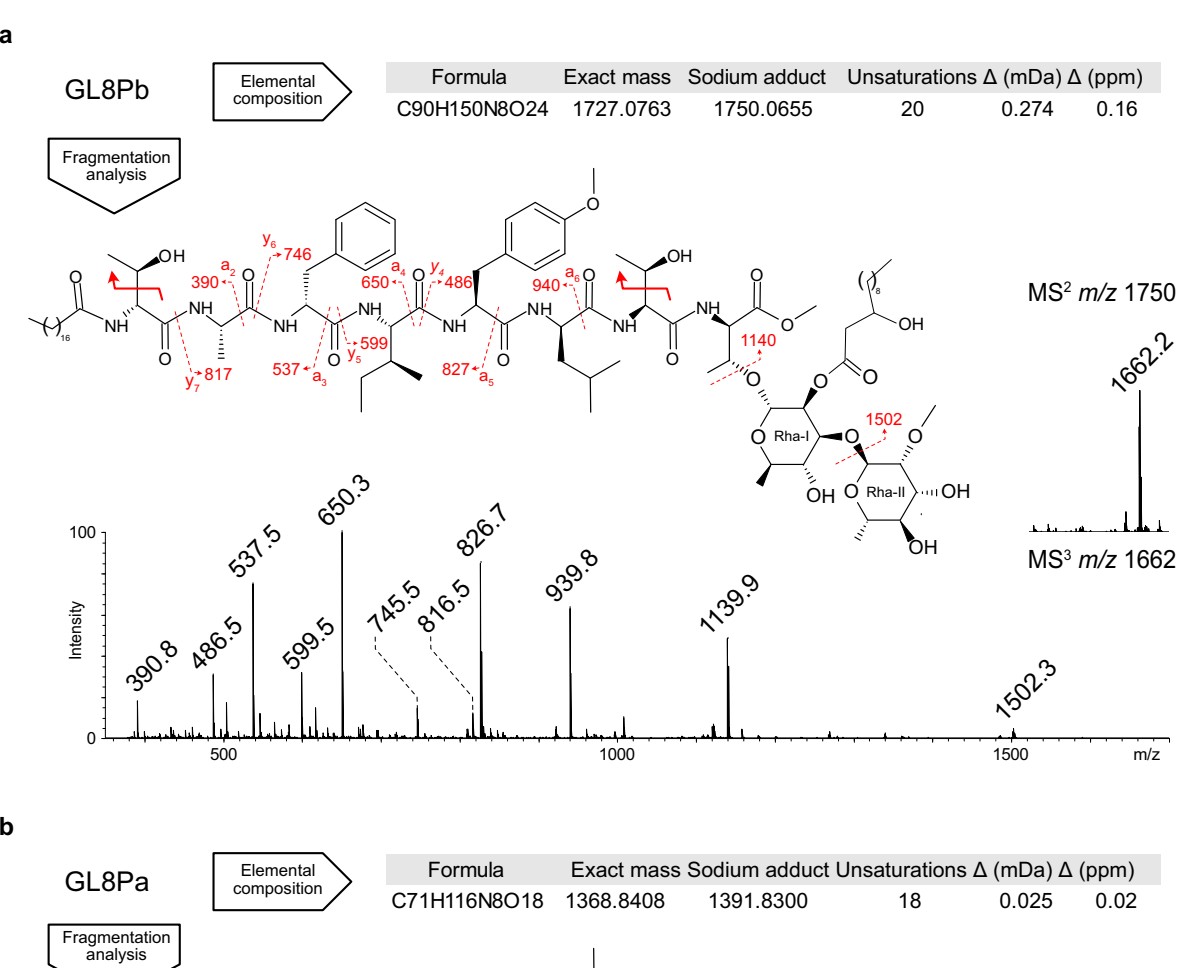

| Formula | Exact mass | Sodium adduct | Unsaturations | Δ (mDa) | Δ (ppm) |
|---|---|---|---|---|---|
| C90H150N8O24 | 1727.0763 | 1750.0655 | 20 | 0.274 | 0.16 |

**b**

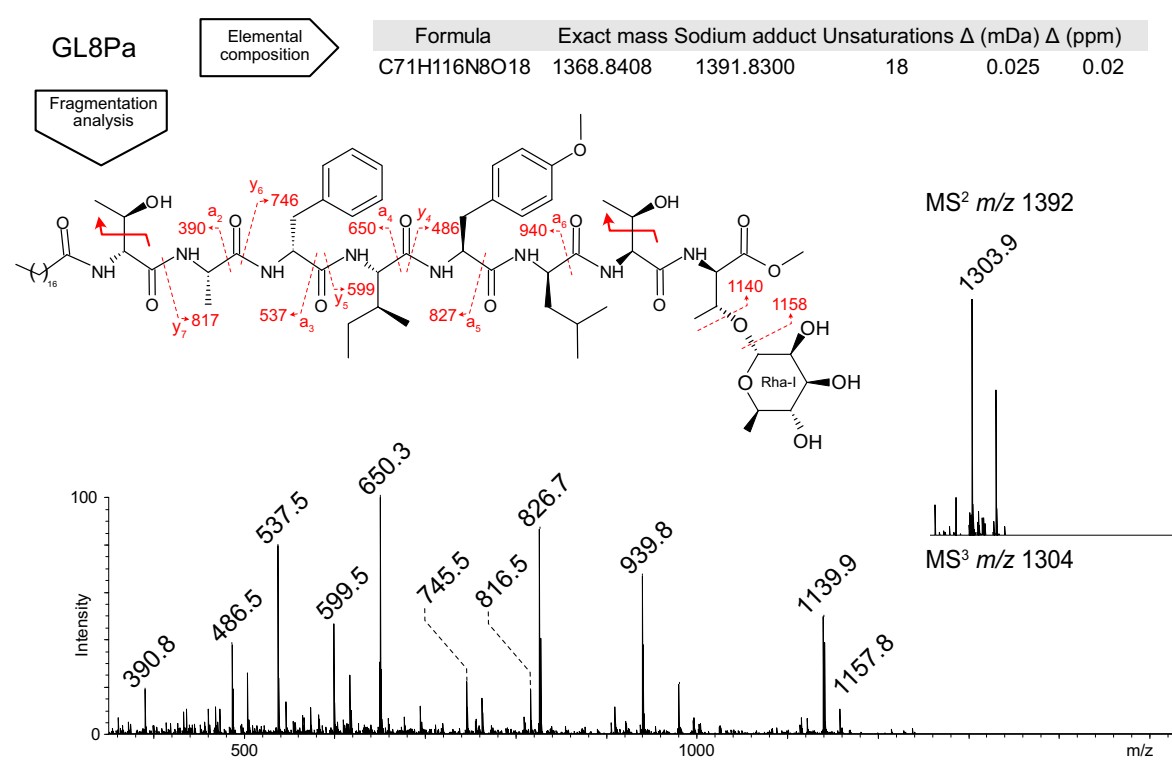

| Formula | Exact mass | Sodium adduct | Unsaturations | Δ (mDa) | Δ (ppm) |
|---|---|---|---|---|---|
| C71H116N8O18 | 1368.8408 | 1391.8300 | 18 | 0.025 | 0.02 |

compared to those infected with either R or RΔ*4690c::4690c*, but a burden equivalent to that observed with the *M. abscessus* S variant, as judged by the CFU counts at 2- and 4-day post-infection (dpi) (Fig. 6h). These observations in zebrafish mirror the ex vivo findings in THP-1 cells, further substantiating the critical role of GL8P in facilitating *M. abscessus* interactions with host cells and in promoting sustained intracellular multiplication.

These findings support the relevance of GL8P in host-pathogen interactions and point toward its role in more complex animal models.

### RΔ*4690c* is attenuated in mice

To assess GL8P's role in *M. abscessus* virulence, R, RΔ*4690c*, and RΔ*4690c::4690c* were injected intravenously into C3HeB/FeJ mice[37]. In agreement with a previous report, the R strain exhibited a hyper-

**Fig. 5 | Glycan and lipid variations between GL8Pb and GL8Pa. a** Elemental composition, $C_{90}H_{150}N_8O_{24}$ of GL8Pb established by FT-ICR MS analysis with an estimated error of 0.16 ppm (Δ) (Fig. S6). The eight nitrogen atoms and twenty unsaturations are in accordance with the amino acids and monosaccharides identified by NMR. The MS$^2$ fragmentation of the parent ion at $m/z$ 1750 generated [M−88+Na]$^+$ fragment at $m/z$ 1662 due to the loss of the two free threonine side chains (red arrows). The MS$^3$ fragmentation pattern of the parent ion at $m/z$ 1662 was dominated by a$_2$ to a$_6$ fragments as well as y$_4$ to y$_7$ (red dashed arrows indicate fragmentation according to nomenclature), confirming the peptide sequence established by NMR. The loss of the terminal 2-*O*-methyl rhamnose was observed owing to the [M−160+Na]$^+$ fragment at $m/z$ 1502, while the ion at $m/z$ 1140 suggests

that a hydroxylauric acid substitutes the 2-*O*-methyl-α-rhamnopyranosyl(1-3)2-*O*-acyl-α-rhamnopyranose glycan motif. **b** Elemental composition of GL8Pa $C_{71}H_{116}N_8O_{18}$ was established by FT-ICR MS analysis with an estimated error of 0.02 ppm (Δ) (Fig. S6). Compared with GL8Pb, the difference of atomic composition and the decrease of two insaturations suggest a modification of the glycan moiety. The MS$^2$ fragmentation of the parent ion at $m/z$ 1392 similarly produced [M−88+Na]$^+$ fragment at $m/z$ 1304 (red arrows). Its MS$^3$ fragmentation pattern was dominated by a$_2$ to a$_6$ fragments as well as y$_4$ to y$_7$ (red dashed arrows), suggesting a common peptide core with GL8Pb. Fragment ions [M−164+Na]$^+$ at $m/z$ 1140 and [M−146+Na]$^+$ at $m/z$ 1158 demonstrated the presence of an unmodified terminal rhamnose (Source data https://doi.org/10.5281/zenodo.14918824).

lethality phenotype in mice[21]. Strikingly, all mice infected with RΔ*4690c* survived at 20 dpi, while all animals infected with RΔ*4690c::4690c* succumbed rapidly, similar to the parental R strain (Fig. 7a). Additionally, the hypo-virulent phenotype of Δ*MAB_4691c* was further confirmed in BALB/c mice (Fig. 7b), along with the absence of GL8P in the polar lipid fraction of this mutant (Fig. S7a). This implies that loss of GL8P production correlates with increased survival in mice. To confirm RΔ*4690c*'s attenuated phenotype, CFU were determined in different organs of infected C3HeB/FeJ mice for up to 30 dpi. A significant decrease in RΔ*4690c* loads was observed in the lungs and kidneys at 15 dpi (Fig. 7c, d) and in the kidneys and spleen at 30 dpi (Fig. 7d–f) compared to the organs from R-infected mice. Complementation restored WT CFU levels in the spleen (Fig. 7f). Similar CFU decrease in the lungs, liver, and spleen was observed at 30 dpi in RΔ*4690c*-infected BALB/c mice (Fig. S7b–d). These results support the role of GL8P in maintaining *M. abscessus* infection in mice.

### GL8Pb is recognized by sera from *M. abscessus*-infected CF patients

The serological response to GL8Pb was assessed using sera from CF patients infected with *M. abscessus*[38,39]. Sixteen positive patients, selected based on their positive response to two *M. abscessus*-derived antigens (phospholipase C and the TLR2eF extract)[38], and their positive *M. abscessus* cultures, were compared with a group of sixteen patients negative in both serology and culture. A GL8Pb-specific ELISA showed that 11 out of 16 positive sera (69% sensitivity) tested positive for anti-GL8Pb IgG production (Fig. 8), while only 1 out of 16 negative sera displayed a positive IgG response against GL8Pb (94% specificity). These results indicate that GL8Pb is recognized by sera from *M. abscessus*-infected CF patients. However, further confirmation and expansion of these results are required before considering GL8P as a reliable diagnostic marker for *M. abscessus* infections.

## Discussion

The S-to-R transition is a crucial aspect of the pathophysiology induced by *M. abscessus*, often associated with severe infectious manifestations across various cell types, animal models, and human cases[6–8,19–21]. Comparable morphotype shifts have been noted in other strict and opportunistic species, including *Mycobacterium canettii*[40], *Mycobacterium marinum*[41,42], *Mycobacterium avium*[43], and *Mycobacterium kansasii*[44], wherein S forms typically exhibit lower virulence as compared to their R counterparts. In these instances, the morphological switch entails the irreversible loss of major cell surface lipids, like GPL or lipooligosaccharides[33,40,44]. In *M. canettii*, spontaneous variants with mutations in the polyketide-synthase-encoding *pks5* locus show a defect in lipooligosaccharide production and morphotype alterations[40]. These R variants may arise due to evolutionary pressures, favoring a more virulent or persistent phenotype in cellular and animal infection models. In hosts infected by *M. abscessus*, the robust antibody response against GPL[27] likely exerts a strong immune pressure, triggering the S-to-R transition as a mechanism of immune evasion, favoring the proliferation of bacterial forms lacking surface GPLs[45]. These observations prompted us to further investigate how the bacilli

adapt within the host following the loss of GPL, which represent a major component of the cell wall[27]. Through in silico and structural analyses, we demonstrate here the synthesis of two variants of a novel GPL-related lipid, GL8Pa, and GL8Pb, crucial for maintaining the virulence of the R form, as evidenced in cellular and animal models. However, we acknowledge that the amount of GL8P is considerably lower than GPL, likely contributing to its oversight in previous studies. Moreover, most published studies focused on the S variant of *M. abscessus*, where GL8P is largely hidden by the abundant GPL pool during TLC separation of polar lipids. In contrast, in the R variant lacking GPLs, GL8P is more easily detectable, as shown in this study. Strikingly, GL8P is synthesized in vivo, as evidenced by the robust anti-GL8Pb humoral response detected in *M. abscessus* patients.

Despite being produced in low amounts (at least in vitro), the loss of GL8P synthesis due to *MAB_4690c* gene disruption is crucial for *M. abscessus* infection, as revealed by the attenuated phenotype in THP-1 cells, zebrafish, and mice. We acknowledge that while only THP-1-cells were assessed here as a cellular model, further studies will extend this work to primary human macrophages. However, the fact that we observed similar results in vivo, in zebrafish macrophages, lends confidence to the phenotype observed. Unlike zebrafish embryos possessing only an innate immune system, mice, having both innate and adaptive immunity, can resist to killing by Δ*MAB_4690c* or Δ*MAB_4691c*. This emphasizes the importance of adaptive immunity, as shown previously for *M. abscessus* S and R, unlike *M. chelonae*[21]. Interestingly, *MAB_4690c* also shares strong homology with *pstA* from *M. avium* subspecies *paratuberculosis* (MAP), the causative agent of Johne's disease in ruminants[46,47]. This analogy with MAP is based on the description of a PstA-derived lipid involved in MAP biofilm formation, virulence, and immunogenicity[46,47]. Moreover, the attenuated phenotype of S Δ*MAB_4691c* in mice aligns with a strong defect in intra-amoebal and intra-THP-1 cells growth/survival of a transposon mutant in *MAB_4691c*[48].

To our knowledge, R Δ*MAB_4690c* is the second example of an avirulent R mutant, as deletion of *MAB_4780*, encoding a dehydratase participating in mycolic acid metabolism, resulted in high attenuation in THP-1 cells, amoebae, and zebrafish due to the incapacity to produce cords[49]. It is noteworthy that R Δ*MAB_4690c* is not defective in cording, suggesting distinct attenuation mechanisms in R Δ*MAB_4780* and R Δ*MAB_4690c*. Interestingly, a previous *M. abscessus*-S/R RNA-seq study[33] revealed a slight change in the expression level of *MAB_4690c* in the R compared to the S strain, further validating the importance of GL8P in the behavior of *M. abscessus* R inside the host.

In mycobacteria, complex lipids exhibit wide structural diversity and biological functions[27,30,50,51]. Our structural determination identified two lipidic entities, revealing that GL8Pb is built on a TAFIYLTT octapeptide with an *O*-methylated tyrosine. This sequence closely resembles the TAFIYTTT inferred from bioinformatic predictions, suggesting the potency of in silico methods in identifying and predicting additional GPL-related components in mycobacteria. Interestingly, a lipooctapeptide has been described in an R variant of *M. avium*[52], displaying a different peptidic core sequence from GL8P but

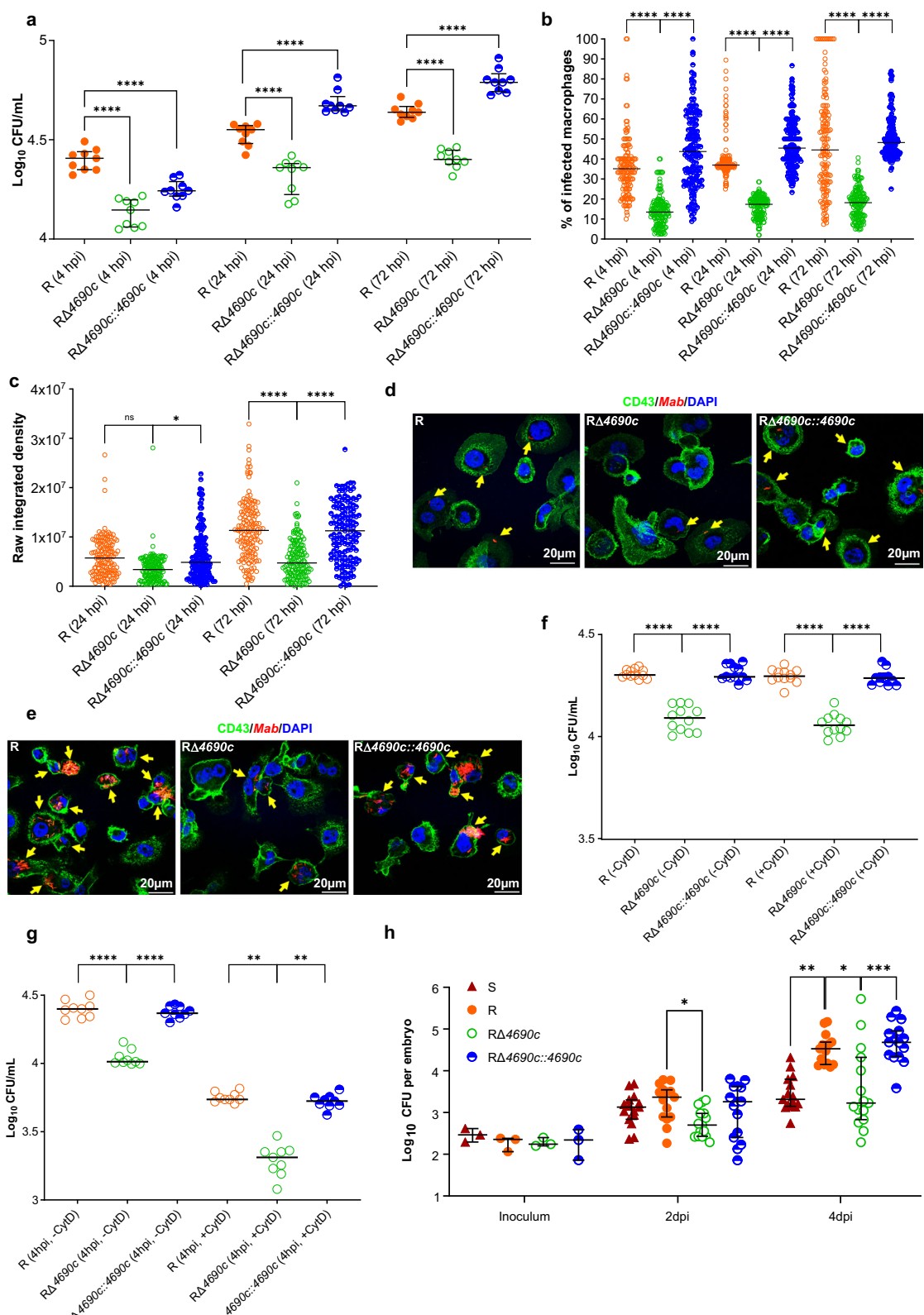

consistent with in silico predictions from the *M. avium* complex genomes (Data S1).

The glycan moiety linked via an *O*-glycosidic bond to the β position of Thr-8 comprises two α-rhamnopyranosyl residues (I) and (II) connected by an α 1-3 glycosidic bond. Both are modified at the C2 position, with the former holding a putative hydroxylated lauric acid and the latter a methoxy group. The lipid portion also includes a

stearyl substituent at the *N*-terminal end. Considering the importance of rhamnose-containing antigens in bacterial virulence and growth, such as the Lancefield antigens from *Streptococcus*[53], it is noteworthy that, similar to previously described GPLs from *M. abscessus*[30], *M. xenopi* or *M. peregrinum*, GL8Pb also possesses a di-*O*-rhamnosyl substituent. The presence of rhamnose in GL8P raises questions about its biological significance, given the structural relatedness with

**Fig. 6 | Reduced adhesion and survival of Δ*MAB_4690c* in THP-1 cells and zebrafish embryos. a** Intracellular bacterial burden was reduced in THP-1 cells infected with Δ*MAB_4690c*. CFUs were quantified at 4, 24, and 72 pi. Data represent mean values ± SD from three independent experiments in triplicate (n = 9). One-tailed Mann-Whitney test, ****p < 0.0001. **b** The percentage of infected cells was lower with RΔ*4690c* infection at 4, 24, and 72 hpi. Data represent mean values ± SD for three experiments (n = 150). One-tailed Mann-Whitney test: ****p < 0.0001. **c** Quantification of raw integrated density for *M. abscessus* strains in infected cells, showing bacterial growth of R, RΔ*4690c*, and *MAB_4690c::4690c* at 24 and 72 hpi. Data represent mean ± SD from three experiments (n = 150). One-tailed Mann-Whitney test: ns non-significant; *p = 0.0146; ***p < 0.0001. **d**, **e** Immunofluorescence images at 4 and 72 hpi (40× magnification) depicting infected THP-1 cells. Bacteria appear in red, nuclei in blue, and CD43, a plasma membrane protein, in green. Yellow arrows highlight *M. abscessus* within cells. **f** Adhesion of bacterial strains to THP-1 cells, ±cytochalasin D, assessed by CFU quantification. Cells were pre-cooled to inhibit bacterial internalization, then exposed to bacterial strains (MOI 100:1) for 1 h at 4 °C. Data represent mean ± SD from three experiments in triplicate (n = 9). One-tailed Mann-Whitney test: ****p < 0.0001. **g** Effect of cytochalasin D on bacterial phagocytosis by THP-1 cells after 4 hrs. Cytochalasin D was added before bacterial exposure and maintained throughout. Bacterial uptake was determined by plating lysates and counting CFUs. Data represent mean ± SD from three experiments in triplicate (n = 9). One-tailed Mann-Whitney test: **p = 0.0025 (R vs. RΔ*4690c*); 0.0084 (RΔ*4690c* vs. RΔ*4690c::4690c*); ****p < 0.0001. **h** Comparative bacterial growth in zebrafish embryos. Embryos were infected with 300 CFU of *M. abscessus* R, S, RΔ*4690c*, or RΔ*4690c::4690c* and compared to mock-infected controls (n = 45 per group). CFUs per embryo were Log10-transformed. *p = 0.0246 (2 dpi); 0.0104 (4 dpi); **p = 0.0018; ***p = 0.0004. Values represent mean ± SD from two experiments. Statistical analysis: two-tailed Kruskal-Wallis test with Dunn's correction. Source Data are provided as a Source Data file.

mannose, a known ligand for lectin domain receptors, like the mannose receptor[30,54]. This suggests a potential implication of the di-*O*-rhamnosyl substituent of GL8Pb in host-pathogen interactions, consistent with the reduced uptake of Δ*MAB_4690c* by THP-1 cells.

In conclusion, this study underscores the high adaptability of *M. abscessus*, akin to strictly pathogenic mycobacteria, through the production of unique GPL-related cell wall components that participate to maintaining high levels of infection in the R form after morphotype transition. The recognition of GL8P by the immune system of the infected host supports its synthesis during host infection. Therefore, while produced in a very restricted panel of NTMs, GL8P represents a potential biomarker to be considered for future diagnostic developments in *M. abscessus* infections, as exemplified by the MAP lipopentapeptide (L5P) for the diagnosis of MAP infection in cattle[55,56].

## Methods

### Ethics statement

Animal experiments were performed according to institutional and national ethical guidelines. Concerning Zebrafish (*Danio rerio*) experiments, animals were kept and handled in compliance with the guidelines of the European Union for handling laboratory animals and approved by the Direction Sanitaire et Vétérinaire de l'Hérault for the ZEFIX-CRBM zebrafish facility (Montpellier) (registration number C-34-172-39). All experiments were approved by the French Ministry of Higher Education, Research and Innovation (MESRI) under the reference APAFIS#24406-2020022815234677 V3. Regarding mice experiments, they were approved by the animal experimentation ethical committee (N°047, Comité d'éthique Pelvipharm, Montigny-le-Bretonneux, France, with agreement A783223) and MESRI under APAFIS #44601-2023090514417206 V3 and APAFIS#11465-2016111417574906 V4. We have complied with all relevant ethical regulations use for animal use. Furthermore, during the course of the experiments, the compliance with the application of these ethical standard was controlled by the animal welfare structure of the animal facilities platform.

Sera were issued from the French National Cohort[38] infected or not with *M. abscessus*.

### Bacterial strains, plasmids, and cloning

Bacteria were cultured in Luria-Bertani (LB) broth at 37 °C. Ampicillin (100 μg/mL), kanamycin (50 μg/mL), and zeocin (25 μg/mL for *E. coli*; 50 μg/mL for mycobacteria) were added when necessary. Restriction endonucleases and modification enzymes (New England Biolabs) were used following the manufacturer's instructions. PCRs were performed using DyNazyme DNA polymerase (Finnzymes). Cycling conditions were as follows: 1 cycle of 5 min at 94 °C, 30 cycles of 20 s at 94 °C, 20 s at 55 °C, and 40 s/kb at 72 °C with a final extension of 10 min at 72 °C. *E. coli* DH5α was used for cloning. To obtain pMC8 and pMC10 (pBSK with Zeo^R flanking the upstream and downstream regions of the deleted segment of *MAB_4690c* or *MAB_4691c*, respectively), pBSK was initially digested by HindIII and EcoRI or HindIII and SpeI, respectively, and then ligated with two PCR products. For pMC8, PCR1 (upstream region) with primers MC16 (5′-CG**AAGCTT**ctgcggacgatcacg-3′) and MC17 (5′-GTGCTCGA**GCTAGC**Gccagcgccatctcac-3′), restricted by HindIII and NheI (restriction sequences are in bold capital letters; additional nucleotides in capital letters), and PCR2 (zeocin resistance gene) with primers MC17b (5′-gtgagatggcgctgg**CGCTAGC**TCGAGCAC-3′)/MC18 (5′-ggcgctgacgatctcGatccccgg**GAATTC**-3′) on pLYG204, restricted with NheI and EcoRI, were used. The downstream region PCR with MC19 (5′-**GAATTC**ccggggatCgagatcgtcagcgcc-3′) and MC20 (5′-CG**TCTAGA**cagaccggctccgtg-3′) digested by EcoRI and XbaI was ligated with the previous plasmid digested with the same enzyme, resulting in pMC8. Similarly, for pMC10, PCR1 (upstream region) with primers MC21 (5′-CG**AAGCTT**gtgatcgacgcgttg-3′) and MC22 (5′-GTGCTCGA**GCTAGC**Ggcgtgctcattgtcg-3′), restricted by HindIII and NheI, and PCR2 (Zeocin resistance gene) with MC22b (5′-cgacaatgagcacgc**CGCTAGC**TCGAGCAC-3′)/MC27 (5′-CG**ACTAGT**Gatccccgggaattc-3′) on pLYG204, restricted with NheI and SpeI, were employed. The downstream region (PCR with MC25 (5′-CG**TCTAGA**gatacgcggtgtggtc-3′) and MC26 (5′-CG**ACTAGT**gtcgccatgtcaacg-3′)) digested by XbaI and SpeI was ligated with the previous plasmid digested with the same enzyme, yielding pMC10.

The rough (R) variant of *M. abscessus* CIP104536^T (Table S1) was cultured in Middlebrook 7H9 broth (BD Difco), supplemented with 0.05% Tween 80 and 10% oleic acid, albumin, dextrose, catalase (OADC enrichment; BD Difco) (7H9^OADC) (7H9-OADC), or on Middlebrook 7H10^OADC (7H10-OADC) at 37 °C, supplemented with antibiotics when necessary. Electrocompetent mycobacteria were transformed using a Bio-Rad Gene Pulser (25 μF, 2500 V, 800 Ω). Clones were selected with 1 mg/mL hygromycin for strains carrying pTEC27 (Addgene, plasmid 30182)[57], enabling tdTomato expression, or 250 μg/mL kanamycin for complementation with pMV361-*MAB_4690c*.

### Construction of *M. abscessus* mutants and complementation

Due to the length of *MAB_4690c* and *MAB_4691c* (8 151 bp and 24 327 bp, respectively), only a portion of each gene (nt 3027 to nt 4255 and nt 8012 to nt 8717, respectively) was deleted. Allelic exchange subtracts were amplified by PCR from pMC8 and pMC9 with primers MC16 and MC20 or MC21 and MC26, respectively. The PCR product was then electroporated into *M. abscessus* with pJV53[35]. Following a double crossing-over event, a portion of the gene was replaced by a zeocin resistance cassette. After sub-culturing in the absence of selective marker, pJV53 was cured. Proper gene disruption was confirmed by PCR using primers MC34 (5′-tgtcaacaccttggtactgc-3′) /MC35 (5′-ccacatatccatagcccagt-3′) and MC36 (5′-gaggccatcgacgagata-3′) /MC37 (5′-tcgtcgagcagatctacaac-3′) for *MAB_4690c* and *MAB_4691c*, respectively. The unique insertion of the zeocin cassette was verified by Southern blotting following restriction with ClaI and NheI and

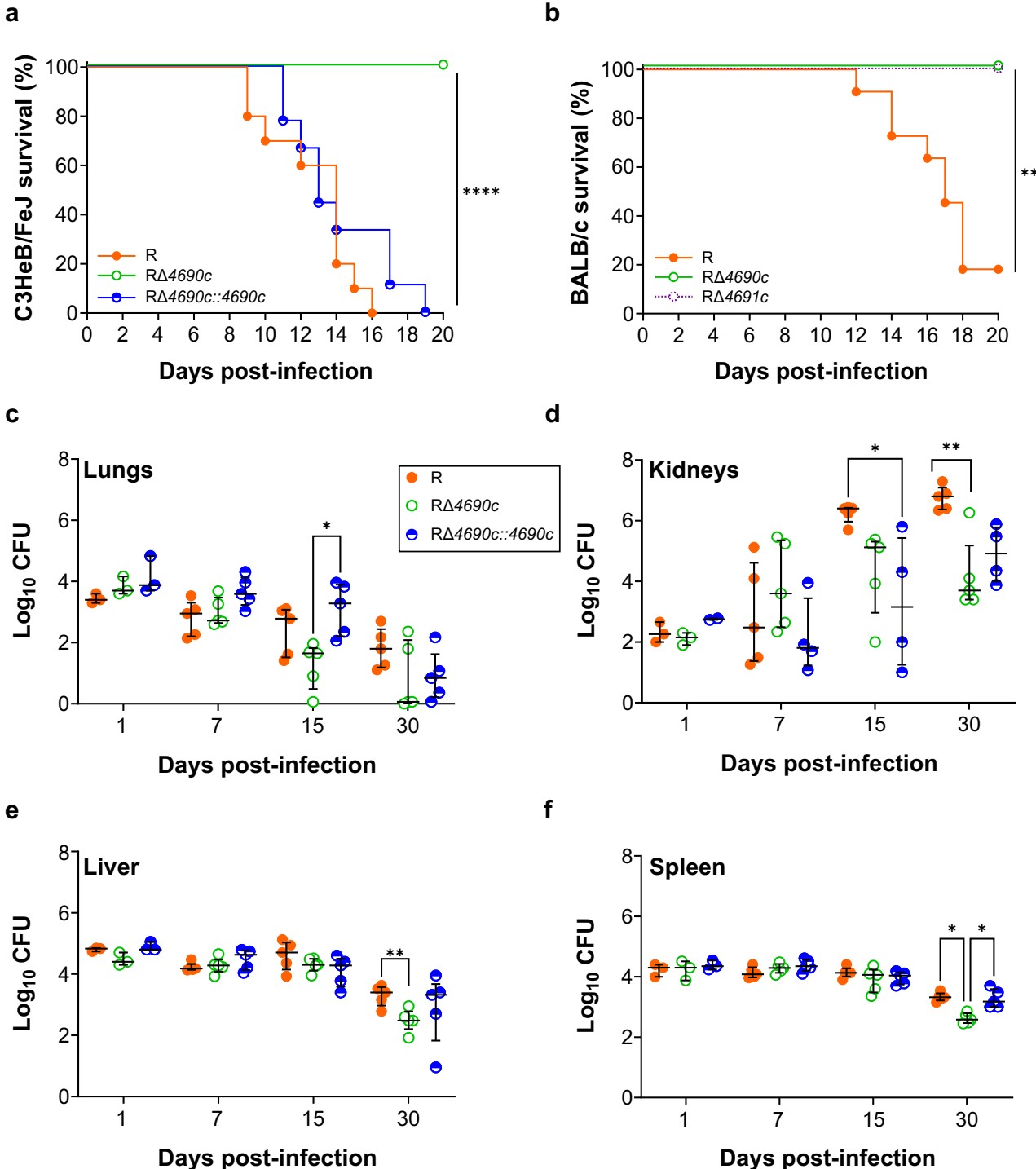

**Fig. 7 | Δ*MAB_4690c* is hypo-virulent in mice. a** Survival of C3HeB/FeJ mice infected intravenously (10⁸ CFU/animal) with R, RΔ*4690c* and *MAB_4690c::4690c* (n = 11). p < 0.0001 (R vs. RΔ*4690c*) and p < 0.0001 (RΔ*4690c* vs. RΔ*4690c::4690c*). **b** Survival of BALB/c mice infected intravenously (10⁸ CFU/ animal) with R and RΔ*4691c* (11 mice per group). p = 0.0001 (R vs. RΔ*4690c*) and p = 0.0001 (R vs. RΔ*4691c*). **c**–**f** Bacterial loads in organs of C3HeB/FeJ mice infected intravenously (10⁶ CFU/animal) with R, RΔ*4690c* and *MAB_4690c::4690c*. Mice were sacrificed at days 1, 7, 15, and 30. Lungs (**c**), kidneys (**d**), liver (**e**), and spleen (**f**) were collected, diluted, and cultured on VCA3 agar plates (n = 5). CFU were counted after 5 days. Results are expressed as median with interquartile range. Each dot in the scatter plots represents an individual mice. At 15 dpi, p = 0.0216 (RΔ*4690c* vs. RΔ*4690c::4690c*) in lungs; at 15 dpi, p = 0.0314 (R vs. RΔ*4690c::4690c*) and at 30 dpi, p = 0.0074 (R vs. RΔ*4690c*) in kidneys; p = 0.0067 (R vs. RΔ*4690c*) at 30 dpi in liver, p = 0.0139 (R vs. RΔ*4690c*) and p = 0.0397 (RΔ*4690c* vs. RΔ*4690c::4690c*) at 30 dpi in spleen. P values were determined using the log-rank (Mantel-Cox) test for Kaplan-Meier survival curves to assess survival statistics significance (**a**, **b**) and by two-tailed unpaired t-test (**c**–**f**) using GraphPad prism program; ns, not significant; *p < 0.05; **p < 0.01; ***p < 0.001; ****p < 0.0001. Source Data are provided as a Source Data file.

labeling with a probe corresponding to the zeocin sequence amplified by PCR (with primers MC17b/MC18 on pLYG204). For complementation, *MAB_4690c* was amplified from genomic DNA using the LongAmp *Taq* DNA Polymerase from New England Biolabs (MA, USA). The resulting 8177 bp amplicon was obtained with primers 5'-GGAATTC**TTCGAA**gtgaaggccgatgcgacc-3' and 5'-GGAATTC**GTTAA C**tcaccgcgctgaggtgct-3' for cloning into the pMV361 after digestion with *Bst*BI and *Hpa*I. pMV361 contains the *hsp60* promoter upstream of

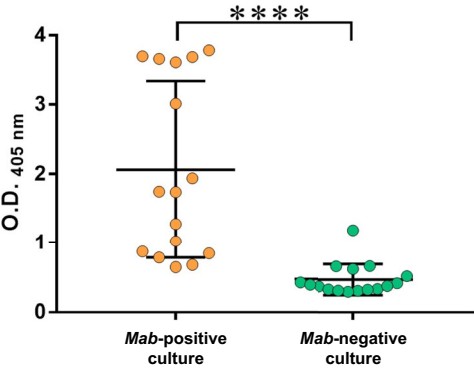

**Fig. 8 | Immunological response against GL8Pb.** The total IgG response was measured in sera from culture-positive (n = 16) and culture-negative (n = 16) CF patients using purified GL8Pb as the coating antigen. Each dot in the scatter plots represents an individual patient and are presented with mean ± SD. *P* values were determined using a two-tailed unpaired *t*-test. ****p < 0.0001. Source Data are provided as a Source Data file.

the cloning site and a kanamycin resistance cassette for selection in the zeocin-resistant mutant.

## Lipid extraction and purification

Lipids extracts were prepared as previously published[58]. After washing with PBS, mycobacteria (n = 3) were freeze-dried, and 50 mg of biomass were resuspended in 2 mL of $CH_3OH$/0.3% NaCl (10:1) and 1 mL of petroleum ether and stirred for 15 min at room temperature. After centrifugation, the upper phase was removed, and the pellet was vortexed in the $CH_3OH$ fraction, and the petroleum ether extraction was repeated twice. Petroleum ether factions were pooled and dried under nitrogen. The apolar lipid fraction was dissolved in 300 µL of $CH_2Cl_2$. The $CH_3OH$ fraction was heated for 5-10 min at 100 °C under reflux. After cooling, 2.3 mL of $CHCl_3$/$CH_3OH$/0.3% NaCl (9:10:3) was added and stirred for 1 h. The mixture was centrifuged for 5 min at 2500 × *g*, and the first supernatant was collected. Finally, 750 µL of $CHCl_3$/$CH_3OH$/0.3% NaCl (5:10:4) was added, and the mixture was stirred for 30 min. After centrifugation, the second supernatant was pooled with the previous one and 1.3 mL of $CHCl_3$ and 1.3 mL of 0.3% NaCl were added in a separating funnel. The separating funnel was stirred vigorously for 5 min, and finally, the lower organic phase was collected after decantation. This polar lipid fraction was dried under nitrogen and dissolved in 300 µL of $CHCl_3$/$CH_3OH$ (2:1). For TLC analysis, 10–20 µL containing 50–150 µg of lipid extract were spotted along a 5 mm strip on a silica gel plate (5–17 µm ALUGRAM Xtra SIL, MACHEREY NAGEL GmbH, Duren, Germany), which was developed in $CHCl_3$/$CH_3OH$/$H_2O$ (90:10:1). After drying, lipids were revealed by spraying with orcinol in 20% $H_2SO_4$ and charring. GL8Pb was purified from the polar lipid extract impregnated on silica. The first purification was done using a 12 g 15 µm silica gel with a column volume of 21.3 mL (CV = 21.3 mL) prepacked column (INTERCHIM, Montluçon, France) that was equilibrated with 100% $CHCl_3$ (7.4 CV). The gradient reached 10% $CH_3OH$ after 14 CV and maintained during 21 CV. After drying 7 mL fractions, they were recovered in $CHCl_3$/$CH_3OH$ (2:1), checked by TLC and MALDI-TOF MS, then pooled. GL8Pb and GL8Pa were eluted together at 10% $CH_3OH$. The second purification was done on a 6 g 50 µm C18 grafted silica (CV = 7.5 mL) prepacked column (INTERCHIM, Montluçon, France) equilibrated with 5 CV of $H_2O$/$CH_3OH$ 0.1% TFA (1:1). The gradient reached 100% $CH_3OH$ 0.1% TFA after 91 CV and maintained for 105 CV. Again, 7 mL fractions were checked by TLC, MALDI-TOF MS, and NMR before further analysis. GL8Pb eluted at 100% $CH_3OH$ 0.1% trifluoroacetic acid (TFA) after GL8Pa. For GL8Pb characterization, only fractions with identical ¹H NMR spectra were pooled resulting in 1 mg material starting from 1.5 g of dry weight bacterial pellet (0.1% yield).

## Nuclear magnetic resonance (NMR)

The samples (n = 2) were repeatedly dried with a mixture of $CDCl_3$/$CD_3OD$ (2:1), or $CDCl_3$/$CD_3OH$ (2:1) to detect hydrogen from the amide group. NMR spectra of the ¹H and ¹³C nuclei were acquired at 293 K using cryo-TCI or TBI probes and AVANCE Neo consoles (BRUKER BIOSPIN GmbH, Rheinstetten, Germany), with resonance frequencies of ¹H at 400, 900 and 1200 MHz. Standard pulse sequences recommended by the manufacturer (zg; selcssfdizs.2; cosygpqf; dipsi2etgpsi; noesygpph; hsqcedetgpsisp2.2; hsqcdietgpsisp2; hmbcetgpl3nd) were used for each homonuclear (¹H/¹H COSY = COrrelation SpectroscopY highlights vicinal protons; ¹H/¹H TOCSY = TOtal Correlation SpectroscopY; ¹H/¹H NOESY = Nuclear Overhauser Effect SpectroscopY highlights spatial proximity of protons less than 5-6 Angström) or heteronuclear experiment (¹H/¹³C HSQC = Heteronuclear Single Quantum Coherence shows proton linked to their carbon; ¹H/¹³C HMBC = Heteronuclear Multiple Bond Correlation highlights quaternary carbon not linked to proton). After acquisition, spectra underwent phase correction and referencing to the residual methanol signal (δ ¹H/δ ¹³C = 3.31/49.29 ppm). Data were analyzed with Topspin 4.0.6 software.

## Gas chromatography coupled to mass spectrometry (GC-MS)

Monosaccharide composition was determined as published previously[30,59,60]. Hydrolysis of 50 µg of sample (n = 1) with 1 µg *myo*-inositol as an internal standard was conducted with 1 mL TFA 4 M for 4 h at 100 °C, followed by drying and desiccation overnight. Reduction was performed for 4 h at room temperature in 500 µL sodium borohydride ($NaBH_4$) 10 mg/mL in 2 M ammonium hydroxide ($NH_4OH$). The reaction was stopped with glacial acetic acid, dried at 55 °C under a $N_2$ stream, co-distilled 5 times with $CH_3OH$ 100%, and finally desiccated overnight. Peracetylation was carried out with 500 µL acetic anhydride for 4 h at 100 °C, and the reaction products were extracted 5 times with $CHCl_3$/$H_2O$ (1:1). The chloroform phase was filtered, dried on sodium sulfate ($Na_2SO_4$), and recovered in 500 µL $CHCl_3$. For GC-MS, 1 µL of itol-acetate derivatives was injected in splitless mode with an automatic sampler on a Solgel 1 MS 30 m × 0.25 mm × 0.25 µm capillary column with the following gradient temperature: 120–230 °C at 3 °C/min, then to 270 °C at 10 °C/min. Compounds were detected after electronic impact at 70 eV on an HP-7820 gas chromatograph coupled to a 5976 single quad under Mass Hunter software (Agilent Technologies, Santa Clara, US) in full-scan mode from 50 Da to 500 Da. Retention times and fragmentation spectra were compared to standard rhamnose and previously characterized 2-*O*-methyl rhamnose.

## Marfey derivatization

After 24 h of hydrolysis at 100 °C with TFA 4 M and drying free amino acids from 0.5 mg of purified GL8Pb (n = 1) or 40 nmol of standard amino acids were derivatized by adding 100 µL Marfey reagent 40 mM in acetone with 20 µL triethylamine (TEA) 1 M in water. The mixture was agitated for 24 h at 37 °C in sealed tubes to avoid evaporation. Then, the reaction was stopped by adding 20 µL HCl 2 M in water, and the dried sample was dissolved in 100 µL dimethylsulfoxide (DMSO). Marfey's derivatives were separated on a Kinetex EVO $C_{18}$ 250 × 4.6 mm 5 µm (PHENOMENEX, USA) at 40 °C with a flow rate of 1 mL/min. A linear binary gradient started from 11:89 $CH_3CN$/$CH_3COONH_4$ 10 mM to 38.5:61.5 in 22 min with UV detection at 340 nm on an Ultimate 3000 HPLC system under Chromeleon 6.8 SR15 software (THERMO FISHER SCIENTIFIC, San Jose, USA). The sample chromatogram was subtracted from matrix reagents to avoid contamination peaks as Marfey residual reagent.

## Mass spectrometry

Before analysis, samples were prepared according to the above protocols (Lipid extraction and purification) without further purification step. For Matrix-Assisted Laser Desorption and Ionisation coupled to Quadrupole Ion Trap Time Of Flight (MALDI Q-IT TOF) MS analysis, the

sample (n = 3) in CHCl3/CH3OH (2:1) was mixed (1:1) with 10 mg/mL 2,5-dihydroxy benzoic acid (DHB) in CHCl3/CH3OH (1:2) as the matrix. One µL was spotted on the target, and MS and $MS^n$ acquisitions were carried out on Axima Resonance under Shimadzu Biotech MALDI-MS 2.9.8.1 software (SHIMADZU, Kyoto, Japan). Laser energy was optimized for each sample between 90 and 130 eV, and acquisition was conducted in positive reflectron mode, from 50 to 3000 Da mass range. External calibration was performed on glucidex polysaccharides covering the same mass range. For $MS^2$ and $MS^3$ analyses, the parent ions were selected with a resolution of 250 and 70 then collided with a voltage between 300 and 600 V.

High-field Fourier transform Ion Cyclotron Resonance (FT-ICR) MS was carried out on SolariX MRMS (BRUKER DALTONICS GmbH, Bremen, Germany), equipped with a 12 T superconducting magnet and a dynamically harmonized ICR cell, in positive ion mode. Before spotting, one volume of sample (n = 1), one volume of PEG2000 (1 mM in water) as an internal standard, and two volumes of matrix were mixed. The laser was set to 25% of power, and the mass range was selected from 150 to 3000 Da. 10–20 scans, each resulting from 200 laser shots, were accumulated for each mass spectra, which were recorded using 4 million points (transient 1.12 s). Electrospray ionization (ESI) was carried out by diluting the sample (1:100) in CH3OH with 1% v/v HCOOH. External calibration with sodium trifluoroacetate (NaTFA) resulted in 10 ppm accuracy, while internal calibration with PEG resulted in a mass accuracy below 0.2 ppm. The crude formula was determined through a query from $C_{0-100}H_{0-200}N_{0-20}O_{0-30}$ with the online tool https://ms.epfl.ch/applications/theoretical-calculations/ using the measured exact mass of the sodium adduct and a maximal error of 1 ppm.

Five to ten µg were injected (n = 1) onto a 150 × 2.1 mm 2.7 µm Halo 90 Å C18 column after 22 min of equilibration with 60% solvent A water/ACN (60:40) 0.1% HCOOH and 40% solvent B isopropanol/ACN (90:10) 0.1% HCOOH at a flow rate of 0.15 mL/min and 50 °C. The gradient reached 100% solvent B at 49 min and was maintained for 8 min. Acquisition was done on Amazon ETD ionic 3D-trap under Compass Hystar software (BRUKER DALTONICS GmbH, Wissembourg, Germany) in positive full-scan mode at 52,000 a.m.u./s. $MS^2$ fragmentation of the three most intense ions was set throughout the chromatogram. A second acquisition dedicated to the detection of the parent ion at m/z 1750 ion was used to compare its relative intensity in polar lipid extracts.

With the exception of FT-ICR, as previously delineated, all data were manually analyzed and curated in accordance with the m/z values deduced from monosaccharides, amino acids, and lipids. All samples were analyzed a minimum of three times to ensure technical reproducibility.

## THP-1 cells infection

THP-1 immortalized monocytic cells (ATCC TIB-202) were used as described[61]. Cells were cultured in RPMI medium supplemented with 10% fetal bovine serum (Sigma-Aldrich) and incubated at 37 °C with 5% CO2. Cells were differentiated into transformed monocytic cell line, THP-1, with 20 ng/mL phorbol myristate acetate in 24-well flat-bottom tissue culture microplates ($10^5$ cells/well) and incubated for 48 h at 37 °C with 5% CO2. Infection with *M. abscessus* strains carrying the pTEC27[57] (MOI 2:1) was conducted for 4 h at 37 °C with 5% CO2. Cells were washed three times with PBS and incubated with RPMI/FBS supplemented with 250 µg/mL amikacin for 2 h to eliminate extracellular bacteria. The amikacin-containing medium was then aspirated, and cells were washed three times with PBS. To assess CFU at 24 h and 72 h post-infection (hpi), infected cells were further incubated in the presence of 50 µg/mL amikacin at 37 °C. At 4, 24, and 72 hpi, THP-1 cells were washed and lysed with 100 µL of 1% Triton X-100. Lysis was stopped by adding 900 µL of PBS, and serial dilutions were plated. CFUs were enumerated after 5 days of incubation at 37 °C.

To assess the impact of Cytochalasin D on bacterial internalization, THP-1 cells were pre-incubated with 4 µM Cytochalasin D for 3 h

prior to infection. Strains carrying pTEC27 were added to the cells at a MOI of 2:1. Cytochalasin D was maintained for the next 2-h incubation with 250 µg/mL amikacin to eliminate extracellular bacteria. At 4 hpi, cells were washed and lysed with 1% Triton X-100, and serial dilutions were plated on LB agar for CFU enumeration.

## Adhesion assays

To assess bacterial adhesion with and without Cytochalasin D treatment, THP-1 cells were grown and differentiated with PMA. For the Cytochalasin D condition, cells were treated with 4 µM Cytochalasin D[62] for 3 hrs prior to the bacterial attachment assay to inhibit actin polymerization and block phagocytosis. In the untreated condition, no Cytochalasin D was applied, allowing cells to maintain normal cytoskeletal function. Before infection, both Cytochalasin D-treated and untreated cells were seeded at a density of 100,000 cells per well in a 24-well plate and placed on ice for 30 min to reduce cellular activity and prevent phagocytosis. Strains were added to each well at a MOI of 100:1, and incubation was allowed to proceed on ice for 30 min. Plates were centrifuged at $1000 \times g$ for 1 min to facilitate bacterial sedimentation onto the cells. Non-adherent bacteria were removed by washing the cells three times with 1 mL of cold 1x PBS[63,64]. To release adherent bacteria, cells were lysed by adding 0.1 mL of 0.1% Triton X-100, diluted in PBS and serial dilutions were plated on LB agar for CFU counting after 4 days of incubation at 37 °C.

## Immunofluorescence staining of infected THP-1 cells

For microscopy-based infectivity assays, THP-1 cells were cultivated on coverslips in 24-well plates at a density of $10^5$ cells/well, incubated for 48 h at 37 °C with 5% CO2. Cells were infected with strains carrying the pTEC27 at an MOI 2:1 for 4 h, washed, treated with amikacin, and fixed at 24 hpi with 4% PFA in PBS for 20 min. Permeabilization was achieved using 0.2% Triton X-100 for 20 min. Following a 20 min blocking step with 2% BSA in PBS supplemented with 0.2% Triton X-100, cells were incubated with anti-CD43 antibody (Becton Dickinson; dilution 1:1000) for 1 h and with an Alexa Fluor 488-conjugated anti-mouse secondary antibody (Molecular Probes, Invitrogen). Cells were stained with 1 µg/mL 4′,6-diamidino-2-phenylindole (DAPI; Becton Dickinson) for 5 min, washed with PBS, mounted onto microscope slides using Immumount (Calbiochem), and examined using a confocal microscope equipped with a 40× objective (Zeiss LSM880).

## Quantification of raw integrated density in fluorescence imaging

Raw integrated density measurements were obtained and analyzed using the Zen imaging software (Zeiss, Oberkochen, Germany). Regions of interest (ROIs) were manually selected to define the areas for quantification, specifically focusing on infected host cells. For each ROI, the raw integrated density, representing the sum of pixel intensities within the selected area of infected cells, was calculated automatically by the software. Background values were subtracted by selecting neighboring non-fluorescent areas. Data were then exported and analyzed statistically using GraphPad Prism software, version 10 (GraphPad Software, San Diego, CA, USA). Datasets were checked for normality, and appropriate statistical tests were applied based on data distribution.

## Zebrafish maintenance and infection

Eggs were obtained by natural spawning, then bleached and incubated at 28.5 °C in Petri dishes containing E3 medium (5 mM NaCl, 0.17 mM KCl, 0.33 mM CaCl2, 0.33 mM MgSO4). All experiments in the current study were performed using the *golden* mutant[20].

Embryos were enzymatically dechorionated at 24 h post-fertilization (hpf) using 1 mg/mL of pronase (stock of 5 mg/mL diluted in E3) and placed in 60 mm Petri dishes containing E3 at 28.5 °C. Microinjection was performed as previously described[65]. Embryos were injected in the caudal vein with ±250 CFUs of either the R, S,

RΔ*MAB_4690c* or RΔ*MAB_4690c::4690c* carrying pTEC27[57] or the PBS control. Injected embryos were rinsed twice, transferred individually in 48-well plates containing E3. Inoculum was checked *a posteriori* by microinjecting a PBS drop and plating onto 7H10[OADC]. Bacterial burden was determined at 2 and 4 dpi, as previously described[65], on LB agar plates supplemented with rifampicin (15 µg/mL) and hygromycin (500 µg/mL).

## Mice infection

Three months-old male and female *Mus musculus* C3HeB/FeJ and BALB/c (Janviers Labs, France) mice used in this study were maintained with free access to food and water according to the European animal welfare regulations The dark/light cycle was with an alternation of 12 h, in an ambient temperature of 21–22 °C and a 40% humidity level. We have complied with all relevant ethical regulations use for animal use. Furthermore, during the course of the experiments, the compliance with the application of these ethical standard was controlled by the animal welfare structure of the animal facilities platform. C3HeB/FeJ and BALB/c mice were challenged intravenously with *M. abscessus* using various doses for the survival study ($10^8$ cells per mice) or the bacterial growth study ($10^6$ cells per mice). Inoculations were performed with fresh aliquots *M. abscessus* strains grown on 7H9 diluted in physiological serum in a total volume of 200 µL per mouse injected in the lateral tail vein[21]. At various time points post-infection, lungs, liver, spleen, and kidneys were collected in sterile distilled water, homogenized and five-fold serial dilutions were then plated on VCA3 plates (Vancomycin, Colimycin, and Amphotericin B, bioMérieux, France) for CFU enumeration. Plates were incubated at 37 °C for 5 days. Results were expressed as the mean $\log_{10}$ CFU per organ. In BALB/c mice, the minimum detection limit per organ was 20 CFU (or 1.3 $\log_{10}$ CFU) per lung, kidney, spleen, and liver. For each time-point, a total of 5–7 mice were infected (three for day 1). Eleven mice per group were included for infection with high bacterial doses.

## GL8Pb ELISA

A total of 32 sera from CF patients, originating from a previously described cohort[38] including 16 patients with *M. abscessus* positive-cultures, and 16 *M. abscessus* negative-cultures were analyzed. IgG levels against GL8Pb were evaluated by indirect ELISA assays using purified GL8Pb coated at 1 µg/mL in carbonate/bicarbonate 0.1 M buffer and incubated with sera diluted at 1/400. ELISA was developed as previously described[38] using alkaline phosphatase-conjugated goat anti-human IgG (Southern biotechnology, Birmingham, USA) diluted in PBS containing 0.05% (v/v) Tween 20 and 0.5% BSA. OD mean ± standard deviation was calculated for each patient group.

## Bioinformatics analyses

Complete Genomic sequences were extracted from GenBank (https://www.ncbi.nlm.nih.gov/genbank/) in January 2024 and biosynthetic gene clusters were identified using the software tools for secondary metabolite biosynthesis analysis antiSMASH 7.0[66]. The NRPS domain analysis and annotation, prediction of the core chemical structure of NRPSs were confirmed using the antiSMASH. Orthologs of genes were analyzed using National Center for Biotechnology Information (NCBI) blast search. For the phylogeny analysis, the genome sequences data were uploaded to the Type (Strain) Genome Server (TYGS), available under https://tygs.dsmz.de, for a whole genome-based taxonomic analysis[67]. For the phylogenomic inference, all pairwise comparisons among the set of genomes were conducted using GBDP and accurate intergenomic distances inferred under the algorithm "trimming" and distance formula d5. 100 distance replicates were calculated each. Digital DDH values and confidence intervals were calculated using the recommended settings of the GGDC 4.0. Results were provided by the TYGS on 2024-03-18.

## Statistical analysis

Data visualization and statistical analysis were executed using Prism 10.2.2 software (Graphpad, USA). All data are presented with median and interquartile range as detailed in the figure legends. The n value denotes either the number of biological replicates or the number of independent experiments (or as otherwise specified) and indicated in the legends. For determination of bacterial counts in THP-1 cells, mice, and zebrafish, CFUs were $\log_{10}$ transformed. The significance between multiple selected groups was determined using either One-way ANOVA with Tukey's multiple comparisons test after validating the normality of the data, or Kruskal-Wallis test with Dunn's multiple comparisons test if the normality of the residuals was disproved. The p values were determined by the log-rank (Mantel-Cox) test for Kaplan-Meier survival curves to assess survival statistics significance in mice. The p-value was considered significant if <0.05, and significance was denoted as follows: *$p < 0.05$, **$p < 0.01$, ***$p < 0.001$, ****$p < 0.0001$.

## Software

Fiji (1.52p) and Zen (black edition) to analyze microscopy images; NMR data were analyzed with Topspin 4.0.6 software. GC/MS were analyzed under Mass Hunter software (Agilent Technologies, Santa Clara, US). Mass spectrometry data were analyzed using Shimadzu Biotech MALDI-MS 2.9.8.1 software (SHIMADZU, Kyoto, Japan). Amazon ETD ionic 3D-trap data were acquired under Compass Hystar software (BRUKER DALTONICS GmbH, Wissemburg, Germany). GraphPad Prism (10.2.2) were used to perform statistical analysis and to create data graphs.

## Reporting summary

Further information on research design is available in the Nature Portfolio Reporting Summary linked to this article.

# Data availability

Source data are provided with this paper. Structural data generated in this study have been deposited in the database under accession code [https://doi.org/10.5281/zenodo.14918824]. Source data are provided with this paper.

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

## Acknowledgements

This project was funded by the French National Research Agency (19-CE15-0012-01; SUNLIVE) and the "Equipe FRM EQU202103012588" funding to L.K. We are grateful to M. Plays, P. Richard, and C. Hamela for their support with zebrafish husbandry. We would like to thank the University of Versailles St Quentin and Inserm for their recurrent funding to J.L.H., the association "Vaincre la Mucoviscidose" for their financial support, and the I2Care Animal Care Centre for maintaining and caring for the animals. We thank Pr. B Marshall (University Hospital and Faculty of Medicine, Southampton, UK) for his thorough review of the manu-script and helpful amendments. We are grateful to the PAGes core facility (Plateformes Lilloises en Biologie et Santé (PLBS)—UAR 2014—US 41) for providing the scientific and technical environment conducive to achieving this work. Financial support from the IRINFRANALYTICS FR2054 for conducting the research is gratefully acknowledged.

## Author contributions

Y.G., L.K., and J.-L.H. set up and supervised the research projects pre-sented in this manuscript. L.D.L., V.L.M., W.D., M.C., B.V., Y.T., X.T., H.L., and I.S.A. carried out the various experiments that led to the results presented in this manuscript. N.D. and F.B. carried out all the bioinfor-matics analysis and wrote their part as well as editing the whole manu-script. L.D.L., V.L.M., W.D., F.B., Y.G., L.K., and J.-L.H. have written and corrected the various versions of the manuscript.

## Competing interests

The authors declare no competing interests.
