## [Transparent Peer Review file · Nature Communications]

A glycosylated lipooctapeptide promotes uptake and growth of *Mycobacterium abscessus* in the host

Corresponding Author: Professor Jean-Louis Herrmann

Version 0:

Reviewer comments:

Reviewer #1

(Remarks to the Author)

Infections caused by *M. abscessus* (Mab) are becoming a serious health problem due to their reported increase in numbers and due to the intrinsic resistance of these opportunistic pathogens to available drugs. Therefore, gaining insights into biology and pathogenicity of Mab is extremely important for the development of novel therapeutic interventions and diagnostic tools. In the submitted work the authors used bioinformatic approach combined with genetic tools to discover a new virulence factor, a glycosylated lipooctapeptide GL8P comprising family of 2 related compounds. They provide thorough chemical characterization of GL8P and prove its role in Mab virulence by state-of-the-art methods. The discovery of GL8P was made in a more virulent rough (R) variant of Mab. Interestingly, R variant of Mab is a result of a loss of related, canonical glycopeptidolipids (GPLs), which is accompanied with the increase in virulence compared to the parent smooth (S) variant. On the contrary, loss of GL8P leads to decreased virulence pointing out to the specific roles of different cell envelope components in pathogenicity of Mab.

The described work is technically sound, the conclusions are supported by the data and I consider the outcome of the described research to be highly important and relevant for the field.

My comments are mostly minor and concern primarily technical aspects of specific experiments:

Lines 68 and 265 – I did not find the information on “GPL constitute up to 70% of mycobacterial cell wall lipids” in ref. 26 or 27. I suspect it is not correct also based on (although schematic) representation of Mab cell envelope in DOI: 10.1016/j.tim.2021.06.006

Line 94 – “(MAB_4698 to MAB_4705c)” should be “(MAB_4689 to MAB_4705c)”

Line 101 – “D-Thr-L-Ala-D-Phe-(L-Ile/L-Ala)-L-Tyr-D-Thr-Thr-D-Thr” should be modified (inserted L at penultimate Thr) “D-Thr-L-Ala-D-Phe-(L-Ile/L-Ala)-L-Tyr-D-Thr-L-Thr-D-Thr”

Comments regarding Fig. 1:

A: Including both the numbers and the names of the genes would be more informative – as e.g. in Fig. 1 in ref. 61. The 2 NRPS in *Msmeg* are mislabeled – they are MSMEG_0401 and MSMEG_0402. Color coding for “transport-related genes” and “regulatory genes” is too similar.

C: I would delete 1C – it is redundant – just showing the prediction of the amino acids presented in 1B (but in the position M4 – Ile was chosen without an explanation why)

Legend to Fig. 1: line 762 – I think that PCP is “esterified” not “etherified”.

It would be helpful if the functions of the known gene products were described in the legend. What is mbtH, gap, etc.?

Comment regarding Fig. S1:

I propose to consider renaming those species that are given different names to “*Mycobacterium*” for consistency with Table S1 and with the rest of the manuscript.

Comments regarding Fig. S2:

A: What is the profile of the whole polar lipid fraction in the solvent C/M/W (90:10:1)? It should be shown, since this is the solvent used for monitoring GL8Ps (as shown in the Fig. S3A, E) and they were isolated from the polar lipids fraction. Were

GL8Ps visible in the whole polar lipid fraction? (One would expect it, since their abundance in the polar lipid fraction appears to be higher than that of PIM2 family of lipids – based on Fig. S2C.)

B: PIM2 is not a major form of PIM in mycobacteria, acylated forms are more common; please, correct labeling (and modify the text, line 123-124) and reveal identities of the other spots on TLC plate.

C: The proof (or reference) for labelling m/z 1446 signal as Ac1PIM2 should be included. PIM2 (or rather acylated PIM2) content appears to be comparable by TLC, but intensity of the m/z 1446 signal, labelled as Ac1PIM2 is higher for the deletion strain. Are the scales on y-axis different? Explain, please. (I would expect comparable intensities of Ac1PIM2 in all three strains, based on TLC if aliquot amounts of the lipids were analysed.)

Line 131: “both compounds were isolated of a 3,4 di-O-acyl trehalose” – please, rephrase (correct) the statement

Line 133: “reversed phase flash chromatography led to higher quantity of NMR checked pure GL8Pb than GL8Pa (Fig. S3E-F)” – please, re-phrase.

Comments regarding Fig. S3:

A: What is the identity of the band in the lane F19?

E: What was the MS profile of fraction 65, which was combined with 64 for further analysis? I am asking because of the relative increase of the quantity of an extra (higher) band in this fraction.

Comments regarding Fig. 2:

B. What is meant by the quantities stated at the upper chromatogram and bottom chromatogram? Are these the total amounts of the material used to get the presented profile? If so, it would be better to use the comparable SI units – preferably moles. Why is the amount of GL8P not stated accurately? According to my rough calculation, 1 mg of GL8Pb is about 580 nmols. That would mean that the amounts of the amino acids in the sample is highly underestimated. Is hydrolysis with 4M TFA at 100°C for 24 hrs sufficient to break the peptide bonds? (I am aware of a different way, e.g. as described in ref. 26 – 22 h, 110°C, 6 M HCl.)

D. For better clarity I would keep fragmentation profiles of the two peaks corresponding to 2-O-Me-Rha and Rha at the same place both in Fig. 2, or in Fig. S6.

Comment regarding Fig. S6.

C: The fragment 170.1 is not shown in the structure.

Line 147: Please change “Marfey derivatives” to “Marfey’s derivatives” in this line and also elsewhere in the MS.

Line 172: I believe that the statement “To elucidate the nature of the acyl chains attached to Thr-1 at the C2 position of the internal rhamnose” should be changed to “To elucidate the nature of the acyl chains attached to Thr-1 and at the C2 position of the internal rhamnose”

Line 179: I am aware that in a general structure of GPL, the regions are specified as “lipid, peptide, carbohydrates” (e.g. as in ref. 10). However, GPL or GL8P are the lipids, so the statement “total lipid moiety” appears to me confusing in this context. Please, consider clarification.

Line 202: I believe it should be specified in the main text of the MS, that/if ex vivo and in vivo experiments were performed with the strains transformed with the pTEC27 expressing tdTomato. This information should be stated consistently also in the Methods. Currently, in the Methods line 475 states that Mab strains were carrying pTEC27, and line 495 states that the cells were expressing tdTomato. It should be unified; I believe it is the same thing.

Paragraph starting at line 221:

What was the purpose for using 2 different kinds of mice for experiments shown in Fig. 5 A and Fig. 5 B, i.e. with KO in 4690 and 4691, respectively?

Paragraph starting at line 235:

Is it known if the patients were infected with S or R variants of Mab?

Line 268: “However, we acknowledge that the amount of GL8P is considerably lower than GPL in M. abscessus, likely contributing to its oversight in previous studies.” What was the yield of GL8P (e.g. per mg of dry weight of Mab) and how it compares with GPL amount in Mab?

Lines 270-272: “Moreover, most published studies were conducted using the S variant, where GL8P is completely masked by the GPL pool during conventional TLC separation of polar lipids, further elucidating why GL8P may have been overlooked previously.” In fact, in ref. 30 Ac1PIM2 is shown to be the major lipid in the polar fraction of R form of Mab (Fig. 1D), while in the present study it looks like GL8P is more abundant (Fig. S2C). Therefore, I would change the argument for “overlooking” GL8P.

Line 279: I do not understand this argument, since in ref. 21 the studied strain is said to be “Mycobacterium abscessus (formerly Mycobacterium chelonae subsp. abscessus)”

Line 285: I think that it should be specified/discussed here, that the Tn mutant in MAB_4691 in this study (ref. 49) was in S variant of Mab. The change in virulence in the mutant underlines the role of GL8P both in S and R variants of Mab. Moreover, the change in expression of 4690c in R and S is minimal (line 291), so the conclusion about “the importance of this compound in the pathophysiology of persistent infection by the R morphotype” (lines 292-293) would deserve stronger

justification.

Line 296: "GL8Pb is built on a TAFIYLTT octapeptide with an O-methylated tyrosine". Also Thr-8 is O-methylated (Fig. 2C, S6B, S7C). Why this is not specified here (and elsewhere?).

Line 318 and 319: "GL8P represents a highly specific biomarker to consider for future diagnostic developments in *M. abscessus* infections" this statement should be supported by comparing its selectivity and specificity with other known/used Mab biomarkers.

Line 387: "CHCl₃ phase" – "organic phase" is more accurate

Line 390-391: Please, show the profile, as I mentioned above (comments to Fig. S2)

Line 392-398: The details of purification procedure are missing – which were the volumes of the columns, collected fractions etc...

Line 411: How much of the sample was used for the analysis?

Line 427: "derivation" should be changed to "derivatization"

Line 429: Was TFA dried before addition of Marfey's reagent? The quantities of the GL8P and the standard used for derivatization should be stated also here (not just at the Figure).

Line 780: comparison of "retention times with characterized *Bacillus cereus* samples" – this is not informative. Please, include more details or reference.

Reviewer #2

(Remarks to the Author)

-This study identifies a novel lipooctapeptide from the cell wall of *M. abscessus* which the investigators refer to as GL8Pb which plays a role in the adherence of *M. abscessus* to cells and to which antibody is present in the sera from individuals with cystic fibrosis suggesting the potential for immunoassay to aid in diagnosis of *M. abscessus* infection.

Please address the following points:

1) Under section "Bacterial strains and plasmids" please list all *M. abscessus* strains used in the first paragraph, their derivation and whether they express the rough or smooth phenotype.

2) The data (Figure 4A) showing Mabs CFU in THP1 macrophages need to include an initial timepoint after amikacin is removed from the culture media and the cells have been washed (time 0). The slope of growth over 72 hours appears to be identical for both wild type and the GL8Pb deletion mutant. This is in keeping with the observation in Figure 4C demonstrating significantly less adherence of the GL8Pb deletion mutant to monocytes under cold conditions which inhibit uptake. Thus, the differences observed in Fig 4A likely reflect a significant difference in initial macrophage uptake. A significant difference at time 0 comparing wild type and the GL8Pb deletion mutant would not support the conclusion of reduced intracellular survival of the GL8Pb deletion mutant in macrophages. In the zebrafish model there are significant differences between the growth of wild type and the GL8Pb deletion mutant, but this data does not allow one to determine whether these differences are due to reduced uptake by permissive cells or reduced intracellular survival.

3. Please make clear in Figure 5A and 5B how many mice are in each group. Although mouse numbers are mentioned, it is hard to correlate what is in the Figure legend with the data in the Figures. Similarly, make clear whether the data showing organ CFU are in C3HeB/FeJ or BALB/c mice. Please explain in results or discussion why the survival of wild type mice is significantly reduced at approximately day 18 (Figures 4A,B) compared to the GL8Pb deletion mutant, yet the difference in organ CFU comparing wild type to percentage of mice surviving is not significantly different until Day 30 (Figs C-F). Also, please provide an explanation as to why growth seems to only increase in the kidneys of mice whereas in the liver, spleen and lungs it remains stable and then declines in all bacterial groups near the end of the experiment. Considering these observations, please comment as to the cause of death in infected mice.

4. Please comment and/or note that a limitation in terms of drawing conclusions regarding Mabs mouse infection with the intravenous infection model is that this utilizes a very high inoculum (10⁶) and does not mimic human lung infection. There are now other mouse models of lung infection which are more clinically relevant to human infection (Tuberculosis, Volume 147, July 2024, 102503).

5. Please indicate whether it is known whether rough or smooth (or mixed isolates) were obtained from sputum cultures of patients with CF who had antibody to GL8Pb detected in their serum.

6. The discussion related to antibody to Mabs exerting selective pressure on Mabs in terms of smooth and rough phenotypes needs to be supported by data. Please supply references indicating that antibody to Mabs has been shown to result in bacterial killing, for example by complement fixation. If no data is available, please modify discussion accordingly. It is established that antibody may have negative or positive effects on pathogen survival (Infection and Immunity April 2021 Volume 89 Issue 4 e00054-21).

7. In light of the above considerations, the title - "An exclusive glycosylated lipooctapeptide governs rough *Mycobacterium abscessus* virulence" is confusing and overstates the findings. The use of the word exclusive does not make sense in the

context of the findings. What is being excluded? Do the authors mean novel or unique? In addition, the word governs does not describe the relationship of the lipooctapeptide to *M. abscessus* virulence. Governing implies overarching control. Other bacterial components/products are also involved in *Mabs* virulence and the data supports a role for GL8Pb in cellular uptake.

Reviewer #3

(Remarks to the Author)

Mycobacterium abscessus is a medically important pathogen with established lipoglycan-dependent phenotypes. Here, Leclercq and coauthors identify two genes of unproven function, demonstrate that they encode factors necessary and sufficient for the biosynthesis of previously unknown glycopeptidolipids (GPLs), determine the GPL structures in detail, and demonstrate clear and complemented virulence effects in cellular and in vivo models. Further, a small study demonstrates GPL-specific human immunoglobulin responses, supporting possible species specificity of the glycans identified. The experiments are generally generated with high rigor with complementation of key phenotypes.

Despite overall high enthusiasm for the originality and breadth of work, we have major concerns about certain interpretations and data display. The paper could be improved with a less-chemistry focused data presentation to make the paper comprehensible to physician, bacteriologist, and immunologist readers of *Nature Communications*.

Major

Some discussion of glycan biosynthesis and smooth/rough (S/R) phenotypes as motivating data is reasonable. However, lengthy discussions both before (p. 3) and after the data presentation (p. 12) pose several problems. The sections are redundant, and they set unfulfilled expectations, as the data do not address "intense remodeling" and/or morphotypic change. The discussion of TLR function (line 261) is also a non sequitur. The authors should either discard much of this detail (especially the discussion on p. 12), or include additional data answer the questions raised: what is the yield of these GPLs relative to other GPLs and/or relative to total cell mass? Do deletions in 4690c and 4691c lead to altered morphology?

The cellular and animal model conclusions must be stated more carefully. Not all zebrafish phenotypes are macrophage-dependent, so the unqualified attribution of outcomes to macrophages (line 218) is invalid. THP-1 cells are immortalized monocytic cells (not macrophages). Also, the use of these models is inadequately justified; additional references and discussion are needed. How do these models relate to human disease? If they were chosen based on S/R virulence phenotypes, is there a clear relationship between the S/R morphotype switch and clinical outcomes?

The main figures should show at least some data for every major claim. However, key data showing the effect of gene deletion on glycopeptide production is in Figure S2. In addition, the title states the GPLs are "exclusive" (which should possibly be rephrased as "species specific"): the authors should consider presenting species analyses (Figure S1) as a main figure. Color coding in this figure must also be added.

Conversely, detailed multidimensional NMR data shown in main (Fig. 2) will not be comprehensible to most generalist readers and could move to supplemental. Additionally, certain chemical interpretations that are apparent to the authors will be inaccessible to most readers. Why do epimerases map specifically to the 1, 3, 6, and 8 positions? What are Marfey derivatives? Why does deshielding of H-2 carbon at 80.77 suggest alpha-rhamnopyranose? What are a2 to a6 fragments? What are M-146 and M-164 fragment ions? Authors should remove or define undefined acronyms: COSY, NOESY, HSQC, HMBC, TOCSY, NOE, CFU, hpi, GDBP, C/M/W and many others.

The cooling assay in Fig. 4C is unusual. It is not explained in the results section and is lacking cites to support its use as an adhesion assay. Please consider repeated using cytochalasin D or another literature-supported technique to block phagocytosis (see Toniolo et al. 2023, *EMBO J.* 42:e113490).

Why wasn't the genetic complement used in Balb/c mice?

MINOR

There is very limited discussion of the human immunoglobulin experiment (Figure 6), such as next steps. Also, ELISAs have been highly non-specific for diagnosis overall, yet there is fairly good specificity here. Does this have to do with surface localization or glycans as particularly good immunoglobulin targets?

Various statistical tests are described in methods without explaining which were used for each figure. Figure-specific statistical test descriptions are needed for *Nature* style, but are missing for Figure 5.

In Fig. 4, panels need to appear alphabetical order. Significance indicators in 4D need to be shifted.

Fig. 4B microscopy needs quantitation of organisms. Similarly, Fig. 4A CFU data are used to support a claim of impaired growth at later timepoints; this would require an additional plot showing ratios of CFU counts at 24 and 72 hours vs. 4 hours.

The claims related to Fig. 5C-F could be softened, since the observed CFU differences are mild and intermittent.

OD values in ELISA usually show loss of sensitivity after OD values of 2, yet many values that distinguish the groups are in the 3-4 range. Are these correct? Could better discrimination be obtained by diluting samples?

The strain designations change from figure to figure, and the single letter designations used in the supplemental figures are not sufficiently explicative for the reader.

Reviewer #4

(Remarks to the Author)

Version 1:

Reviewer comments:

Reviewer #1

(Remarks to the Author)

The authors addressed all of the raised issues to my satisfaction. Nevertheless, I still propose a few minor modifications for their consideration.

(i) Although the names tri-O-acylated phosphatidyl-inositol dimannoside for Ac1PIM2, and tetra-O-acylated phosphatidyl-inositol dimannoside for Ac2PIM2 are still widely used in the literature, I consider the name (mono)acylated phosphatidyl-inositol dimannoside for Ac1PIM2 as proposed by Kordulakova et al., 2003 (DOI: 10.1074/jbc.M303639200), and, consequently, di-acylated phosphatidyl-inositol dimannoside for Ac2PIM2, as more accurate. (The name phosphatidyl-inositol is given to the molecule, which already has 2 acyl groups, so tri-O-acylated phosphatidyl-inositol dimannoside would contain 5 acyls, which is not the case.)

(ii) I appreciate correction in labelling the lipids in Fig. 3B, but I did not see the proof of identification of CL, although it is mentioned in the author's reply. I am surprised that other common phospholipids, such as phosphatidyl ethanolamine or phosphatidyl glycerol, which I would expect to migrate between DAT and PIMs (eg. according to Chiaradia et al., 2017, DOI: 10.1038/s41598-017-12718-4; Fig. 7) were not detected in the polar lipid fraction. In the Figure S1, in which characterization of PIM species is provided, the information in the legend is lacking mentioning the R3 substituent. I think that also in the legend to Figure S1, it would be helpful to mention that Man and methylated Man are present in PIM2 species.

(iii) In the new Figure S5C (upper part showing the fragmentation pattern of the molecule), the authors replaced number 129 by 170. My concern was that in the original fragmentation pattern, 170 was missing, yet it was present in the spectrum. Now the situation is opposite – number 170 is in the fragmentation pattern, but 129 is missing there, but present in the spectrum.

Reviewer #2

(Remarks to the Author)

The investigators have adequately addressed Reviewers concerns.

Reviewer #3

(Remarks to the Author)

Comments for the Authors

Leclercq et al. have meaningfully improved upon their technically sound manuscript by reworking figures, adding the requested controls for THP-1 experiments, and addressing some interpretive issues. However, three major interpretative issues remain:

1. The authors chose infection models based on S/R phenotypes, with the key premise that the R form is the more virulent type of *M. abscessus* in humans. Only 2 cited multi-patient studies (Jönsson et al. 2007 & Hedin et al. 2023) posit key conclusions about R vs. S virulence, and these are based on low patient numbers, so we are not convinced that there is a clear relationship between the S/R morphotype switch and clinical outcomes. Unless the authors can cite unambiguous clinical data supporting the increased human-tropic virulence of the R forms of *M. abscessus*, the infection models do not directly inform human "virulence" in natural disease, but are nevertheless useful in vivo phenotypes in defined experimental settings in cells and zebrafish. We therefore ask that the authors avoid reference to GL8P as a "virulence lipid" in the title and throughout the manuscript.

2. This paper has the potential for interdisciplinary appeal to microbiologists, infectious disease physicians, immunologists, and chemists. Yet, the revised version will likely lose readers due to much too much subspecialty jargon and interpretation

that occurs without any explanation to the non-chemist reader, especially in the abstract and lines 132-201. The question even arises as to whether such subspecialty-focused writing will have broad appeal required by Nature Communications editors.

Tangible suggestions: 1) provide a clear definition for every symbol in the acronym GL8P (abstract) and remove the confusing acronym BGC. Previously, we asked questions like "what are Marfey derivatives?" and "what are a2 to a6 fragments?" to elicit changes to lines 132-201. Yet the authors replied to us directly rather than amending the paper on the readers' behalf - answers to these several questions asked in prior review should now be provided in a further revised manuscript.

3. There is widespread agreement regarding major differences between primary tissue 'macrophages' and the 'transformed monocytic cell line, THP-1.' The best way to respond to prior criticisms of conflating these two very different terms would be to show that GPL knockouts also have altered survival in primary macrophages. Lacking this information, the authors might consider a 'limitation of the study' statement regarding the lack of primary macrophage data. The many instances in which THP-1 cells are designated as macrophages (abstract, line 210, 221, 222, other) are substantively inaccurate and should be amended to say "THP-1 cells."

Minor

The description in the Methods of the new "raw integrated density" measure does not specify that it focuses just on infected host cells (as the authors explained to Reviewer 2 in the response letter). They should add this detail so readers can understand exactly what is being compared in Fig. 6C.

Reviewer #4

(Remarks to the Author)

Version 2:

Reviewer comments:

Reviewer #1

(Remarks to the Author)

The authors addressed all issues I had.

Reviewer #1 (Remarks to the Author):

Infections caused by *M. abscessus* (Mab) are becoming a serious health problem due to their reported increase in numbers and due to the intrinsic resistance of these opportunistic pathogens to available drugs. Therefore, gaining insights into biology and pathogenicity of Mab is extremely important for the development of novel therapeutic interventions and diagnostic tools. In the submitted work the authors used bioinformatic approach combined with genetic tools to discover a new virulence factor, a glycosylated lipooctapeptide GL8P comprising family of 2 related compounds. They provide thorough chemical characterization of GL8P and prove its role in Mab virulence by state-of-the-art methods. The discovery of GL8P was made in a more virulent rough (R) variant of Mab. Interestingly, R variant of Mab is a result of a loss of related, canonical glycopeptidolipids (GPLs), which is accompanied with the increase in virulence compared to the parent smooth (S) variant. On the contrary, loss of GL8P leads to decreased virulence pointing out to the specific roles of different cell envelope components in pathogenicity of Mab.

The described work is technically sound, the conclusions are supported by the data and I consider the outcome of the described research to be highly important and relevant for the field.

My comments are mostly minor and concern primarily technical aspects of specific experiments:

Lines 68 and 265 – I did not find the information on “GPL constitute up to 70% of mycobacterial cell wall lipids” in ref. 26 or 27. I suspect it is not correct also based on (although schematic) representation of Mab cell envelope in DOI: 10.1016/j.tim.2021.06.006

We acknowledge that quantifying GPL is challenging and highly strain-specific. Nonetheless, one study (10.1128/jb.144.2.814-822.1980) suggests that GPL constitutes approximately 65-70% of the fibrillar material in *Mycobacterium avium*. To prevent any misunderstanding, we have removed this detail from the manuscript and now describe GPL simply as a major component (line 66 in the revised version).

Line 94 – “(MAB_4698 to MAB_4705c)” should be “(MAB_4689 to MAB_4705c)”

The requested modification has been incorporated as specified, line 91 in the revised version.

Line 101 – “D-Thr-L-Ala-D-Phe-(L-Ile/L-Ala)-L-Tyr-D-Thr-Thr-D-Thr” should be modified (inserted L at penultimate Thr) “D-Thr-L-Ala-D-Phe-(L-Ile/L-Ala)-L-Tyr-D-Thr-L-Thr-D-Thr”

The requested modification has been incorporated as specified, line 98 in the revised version.

Comments regarding Fig. 1:

A: Including both the numbers and the names of the genes would be more informative – as e.g. in Fig. 1 in ref. 61. The 2 NRPS in *Msmeg* are mislabeled – they are MSMEG_0401 and MSMEG_0402. Color coding for “transport-related genes” and “regulatory genes” is too similar.

In this revised version, we have updated Figure 1 as suggested by the reviewer, based on Figure 1 from reference 61. Additionally, we have provided detailed information on all genes within the GPL locus in a new supplementary table, “Table S1”, which includes orthologs, chromosomal locations with a comprehensive list of genes and functions as annotated in the GenBank.

C: I would delete 1C – it is redundant – just showing the prediction of the amino acids presented in 1B (but in the position M4 – Ile was chosen without an explanation why)

In line with the reviewer’s recommendation, we have removed panel 1C from Figure1. In the fourth module of the NRPS (*MAB_4691c*), the prediction yielded two possible amino acid choices. For the “predicted” structure, Ile was selected, as it was the primary output generated by the AntiSMASH bioinformatics tool.

Legend to Fig. 1: line 762 – I think that PCP is “esterified” not “etherified”.

It would be helpful if the functions of the known gene products were be described in the legend. What is mbtH, gap, etc.?

Thank you for pointing out this error. We have modified the legend of Figure 1 accordingly. The acronym **mbtH** refers to *Mycobacterium tuberculosis*, a family of ten proteins (MbtA-J) found in non-ribosomal peptide synthetase clusters responsible for siderophore synthesis. The acronym **gap** stands for **GPLs**

Addressing Protein. All these acronyms are now listed in the **Table S1** in the revised manuscript, which also provides the known functions of these BGC genes.

Comment regarding Fig. S1:

I propose to consider renaming those species that are given different names to “Mycobacterium” for consistency with Table S1 and with the rest of the manuscript.

In accordance with the reviewer’s recommendation, we have revised the manuscript by renaming the species, as listed by NCBI, to “*Mycobacterium*”.

Comments regarding Fig. S2:

A: What is the profile of the whole polar lipid fraction in the solvent C/M/W (90:10:1)? It should be shown, since this is the solvent used for monitoring GL8Ps (as shown in the Fig. S3A, E) and they were isolated from the polar lipids fraction. Were GL8Ps visible in the whole polar lipid fraction? (One would expect it, since their abundance in the polar lipid fraction appears to be higher than that of PIM2 family of lipids – based on Fig. S2C.)

The TLC of the entire polar lipid fraction from R, RΔ4690c, and RΔ4690c::4690c (in the previous Figure S2) has been added to the new Figure 3C. The mobility of GL8Pb is indicated with an arrow. Due to its very low level of expression, which was barely detectable in the wild-type R, we have also included MALDI-TOF (new Figure 3D) and LC-MS spectra (new Figure 3E-F), which show the ions at m/z 1392 and 1750 for R and RΔ4690c::4690c. Regarding the comparison of GL8Pb’s abundance to PIM₂ in the MALDI-TOF spectrum (new Figure 3D), it is more informative to refer to the TLC of the whole polar lipid fraction (new Figure 3B), which demonstrates an equivalent amount of Ac₁PIM₂ and Ac₂PIM₂ across the three strains.

B: PIM2 is not a major form of PIM in mycobacteria, acylated forms are more common; please, correct labeling (and modify the text, line 123-124) and reveal identities of the other spots on TLC plate.

We agree with the reviewer that the acylated forms are the predominant forms of PIM observed. Initially, we simplified this section as PIM was not the main focus of the study. As a result, lines 118-122 (revised version) have been revised to: The tri-*O*-acylated phosphatidyl-inositol dimannoside (Ac₁PIM₂), tetra-*O*-acylated phosphatidyl-inositol dimannoside (Ac₂PIM₂) and trehalose dimycolate (TDM) contents were comparable in all three strains (**Fig. 3A-B**) while a glycolipid at $R_f = 0.4$ appeared more clearly in the polar lipid fraction of RΔ4690c::4690c (**Fig. 3C**). Additionally, we have provided further information on cardiolipin (CL) and di-*O*-acyl trehalose (DAT) in the new Figure 3B.

C: The proof (or reference) for labelling m/z 1446 signal as Ac1PIM2 should be included. PIM2 (or rather acylated PIM2) content appears to be comparable by TLC, but intensity of the m/z 1446 signal, labelled as Ac1PIM2 is higher for the deletion strain. Are the scales on y-axis different? Explain, please. (I would expect comparable intensities of Ac1PIM2 in all three strains, based on TLC if aliquot amounts of the lipids were analyzed.)

The relative intensity of the signal at m/z 1446 falsely appears higher in the mutant strain due to differences in the y-axis scaling. As shown in the three spectra, the analysis is automatically calibrated to the most intense signal, which is GL8Pb at m/z 1750 for the R and RΔ4690c::4690c strains. Since the mutant strain does not synthesize GL8P, the overall intensity of the spectrum is adjusted to the next highest peak. Generally, we do not consider MALDI-MS suitable for precise quantification because the ionization efficiency of each compound varies (PIMs are highly detected in negative ion mode). To obtain accurate quantification, a response curve for each compound would be required, along with an internal standard. The MALDI-MS experiment in Fig 3D should be interpreted as confirmation that the mutant strain is completely defective in the synthesis of GL8P, as no MS signals for m/z 1391 and 1750 are detected. These clarifications have been added to the legend of the new Figure 3.

Regarding the labeling of the ion at m/z 1446 as Ac₁PIM₂, we know that this compound, which consists of two C16 fatty acids and one C19 fatty acid in *M. smegmatis*, produces a [M-H]⁻ adduct in negative ion mode at m/z 1414, as shown by Kordulakova et al. (see Figure 2 from DOI: 10.1021/acsinfecdis.0c00361). Furthermore, Ac₁PIM₂ from *M. abscessus* is known to contain a 2-*O*-methyl mannose, as demonstrated by

Palcekova et al. (see Figure 1 from DOI: 10.1021/acsinfecdis.0c00361). We have confirmed that purified Ac₁PIM₂ (F40) and Ac₂PIM₂ (F47) from a different purification protocol contain equimolar amounts of mannose and 2-O-methyl mannose, along with myo-inositol (Ins) as shown in the figure below.

Fatty acid methyl ester (FAME) analysis also identified three main lipids in Ac₁PIM₂: C14, C16, and C19 fatty acids, as shown below.

Finally, MS² experiments further support our hypothesis that the ion at *m/z* 1446 corresponds to Ac₁PIM₂ containing one C14, one C16, and one C19 fatty acid, while the ion at *m/z* 1474 corresponds to Ac₁PIM₂ with two C16 and one C19 fatty acids. We have included these spectra in the revised manuscript as Fig. S1A-C, and updated the legend of the new Figure 3 to reflect this information.

Line 131: “both compounds were isolated of a 3,4 di-O-acyl trehalose” – please, rephrase (correct) the statement

This was rephrased as: « Both compounds were first separated from the co-extracted 3,4 di-O-acyl trehalose using silica gel flash chromatography, and their purity was determined through nuclear magnetic resonance (NMR) spectroscopy, MALDI-TOF, and gas chromatography-mass spectrometry (GC-MS) analyses”. (now, lines 133 to 137 in the revised version).

Line 133: “reversed phase flash chromatography led to higher quantity of NMR checked pure GL8Pb than GL8Pa (Fig. S3E-F)” – please, re-phrase.

This was rephrased as: “Subsequently, GL8Pa and GL8Pb were separated by reversed-phase flash chromatography, and a sufficient amount of GL8Pb was purified to allow NMR analysis” (now, lines 137 to 139 in the revised version).

Comments regarding Fig. S3:

A: What is the identity of the band in the lane F19?

This band could not be identified. Our hypothesis is that it may represent a tri-*O*-acylated trehalose, although its quantity was too low to get reliable data. Considering it is also present in the mutant strain, it was not further investigated.

E: What was the MS profile of fraction 65, which was combined with 64 for further analysis? I am asking because of the relative increase of the quantity of an extra (higher) band in this fraction.

The spectra from fractions 60, 65, and 70, which were not included in the revised manuscript, show that the ion at m/z 1750 remains highly intense in fractions 65 and 70 (as shown below). We postulate that the second band corresponds to the lipoform described in Figure S7A :

Comments regarding Fig. 2:

B. What is meant by the quantities stated at the upper chromatogram and bottom chromatogram? Are these the total amounts of the material used to get the presented profile? If so, it would be better to use the comparable SI units – preferably moles. Why is the amount of GL8P not stated accurately? According to my rough calculation, 1 mg of GL8Pb is about 580 nmols. That would mean that the amounts of the amino acids in the sample is highly underestimated. Is hydrolysis with 4M TFA at 100oC for 24 hrs sufficient to break the peptide bonds? (I am aware of a different way, e.g. as described in ref. 26 – 22 h, 110oC, 6 M HCl.)

For the upper chromatogram, the amount of GL8Pb was estimated by weighing a purified fraction using a 0.1 mg accurate balance. While we acknowledge that two hydrolysis methods are available, we are cautious about potential racemization when using HCl. In order to evaluate more precisely the quantity of GL8P, we have established the monosaccharide composition by GC-FID and estimated the quantity of GL8P to 290 nmol rather than 580 nm, showing that we had previously over-estimated it. This explains why the recovery appears low compared to accurately weighted pure amino acid standards. This new evaluation was reported in the new Figure 4B.

D. For better clarity I would keep fragmentation profiles of the two peaks corresponding to 2-*O*-Me-Rha and Rha at the same place both in Fig. 2, or in Fig. S6.

As recommended by the reviewer, we retained the chromatogram in the new Figure 4D and added the corresponding EI-MS fragmentation pattern in Figure S5C.

Comment regarding Fig. S6.

C: The fragment 170.1 is not shown in the structure.

In the new Fig. S5C, the fragment at m/z 129 has been replaced by the fragment at m/z 170, and the legend has been modified with a reference to the Complex Carbohydrate Research Center Database, accessible via this link: <https://glygen.ccrcc.uga.edu/ccrc/specdb/ms/pmaa/pframe.html#na>

Line 147: Please change “Marfey derivatives” to “Marfey’s derivatives” in this line and also elsewhere in the MS.

This correction has been made. (Line 152 in the revised version).

Line 172: I believe that the statement “To elucidate the nature of the acyl chains attached to Thr-1 at the C2 position of the internal rhamnose” should be changed to “To elucidate the nature of the acyl chains attached to Thr-1 and at the C2 position of the internal rhamnose”

The change has been made as proposed. (Lines 176-177 in the revised version).

Line 179: I am aware that in a general structure of GPL, the regions are specified as “lipid, peptide, carbohydrates” (e.g. as in ref. 10). However, GPL or GL8P are the lipids, so the statement “total lipid moiety” appears to me confusing in this context. Please, consider clarification.

As proposed, this has been revised to: « The comparison of the atomic composition with the previously determined di-*O*-rhamnosyl octapeptide suggested that the remaining lipid part contains 30 carbons, 58 hydrogens, and 3 oxygens”. (Lines 182-184 in the revised version).

Line 202: I believe it should be specified in the main text of the MS, that/if ex vivo and in vivo experiments were performed with the strains transformed with the pTEC27 expressing tdTomato. This information should be stated consistently also in the Methods. Currently, in the Methods line 475 states that Mab strains were carrying pTEC27, and line 495 states that the cells were expressing tdTomato. It should be unified; I believe it is the same thing.

We have considered this comment to ensure that this information is specified consistently in the Methods section. Specifically, we have unified the references to the strains carrying pTEC27 and the expression of tdTomato to avoid any confusion (see also Table S2).

Paragraph starting at line 221:

What was the purpose for using 2 different kinds of mice for experiments shown in Fig. 5 A and Fig. 5 B, i.e. with KOs in 4690 and 4691, respectively?

The use of two different mouse models for the experiments shown in Figures 7A and 7B was primarily chronological. Each model, with knockouts in 4690c and 4691c, respectively, was developed and tested concurrently. We included both models in the study to strengthen our argument regarding the loss of virulence observed with these mutants. Presenting these results together emphasizes that the loss of virulence is a consistent outcome across separate gene knockouts, thereby reinforcing the robustness of our findings.

Paragraph starting at line 235:

Is it known if the patients were infected with S or R variants of Mab?

The disease typically progresses from the smooth (S) to the rough (R) variant of *M. abscessus*. Studies by Cullen *et al.* (doi: 10.1164/ajrccm.161.2.9903062), Jönsson *et al.* (doi: 10.1128/JCM.02592-06), and Catherinot *et al.* (doi: 10.1128/JCM.01478-08) documented this progression, confirming that colonization and initial infection are associated with the smooth morphotype. This transition from an S to an R morphotype is well described in the literature and is associated with changes in virulence and persistence during the course of infection in animal models and in patients.

Line 268: “However, we acknowledge that the amount of GL8P is considerably lower than GPL in M.

abscessus, likely contributing to its oversight in previous studies.” What was the yield of GL8P (e.g. per mg of dry weight of Mab) and how it compares with GPL amount in Mab?

Of the three replicate purifications performed in our study, the third started with 1.546 g dry weight of *M. abscessus* R, from which we obtained 53.5 mg of polar lipids after extraction. After the initial silica purification, we pooled 27.4 mg of GL8P-containing fractions, finally yielding approximately 1 mg of purified GL8Pb after the second purification step. The estimated yield was approximately 0.1% relative to dry weight. In contrast, for the canonical GPLs, we successfully prepared at least 10 mg of GPL-2a and GPL-3 from a single silica purification step, starting from 600 mg dry *M. abscessus* S in separate experiments. This corresponds to a yield of approximately 1-3% on dry weight. We propose to include the yield of GL8Pb in the Materials and Methods section. However, it should be noted that when purifying GL8P, we prioritized purity over quantity in order to obtain the highest quality structural analysis, thus reducing the final overall yield. Therefore, any direct comparison of amounts should be interpreted with caution.

Lines 270-272: “Moreover, most published studies were conducted using the S variant, where GL8P is completely masked by the GPL pool during conventional TLC separation of polar lipids, further elucidating why GL8P may have been overlooked previously.” In fact, in ref. 30 Ac₁PIM₂ is shown to be the major lipid in the polar fraction of R form of Mab (Fig. 1D), while in the present study it looks like GL8P is more abundant (Fig. S2C). Therefore, I would change the argument for “overlooking” GL8P.

We agree that the argument for "overlooking" GL8P may be unclear given the findings in reference 30, where Ac₁PIM₂ is identified as the major lipid in the R variant of *M. abscessus*. To clarify this point, in our previous work, we focused on the *m/z* range of 800 to 1600 to highlight various glycoforms of the canonical GPLs, and while GL8Pb at *m/z* 1751 was detected, it was not specifically illustrated. As discussed in the manuscript, the choice of ionization mode in MALDI-TOF is critical for obtaining optimal signals for PIMs (in negative mode) versus GPLs (in positive mode), which limits its suitability for global quantification of glycolipids. However, we acknowledge that the previous phrasing may have led to some confusion, particularly regarding the detection of GL8P. To better align with the current findings, we propose the following revised statement: "Moreover, most published studies focused on the S variant of *M. abscessus*, where GL8P is largely hidden by the abundant GPL pool during TLC separation of polar lipids. In contrast, in the R variant lacking GPLs, GL8P is more easily detectable, as shown in this study." (Lines 286-289).

Line 279: I do not understand this argument, since in ref. 21 the studied strain is said to be “Mycobacterium abscessus (formerly Mycobacterium chelonae subsp. abscessus)”

The recognition of *M. abscessus* as a distinct species, in its own right, was first established in the late 1990s starting 2000's (doi: 10.1099/00207713-42-3-337 ; doi: 10.1128/JCM.37.3.852-857.1999 ; doi: 10.1099/ijls.0.63094-0 ; doi: 10.1007/s10096-011-1510-9), which explains why it was previously referred to differently. Additionally, reference 45 specifically compares *M. abscessus sensu stricto* with *M. chelonae sensu stricto*. This study demonstrates that *M. chelonae sensu stricto* is directly eliminated by the innate immune system, whereas *M. abscessus sensu stricto* exhibits resistance to the innate immune response. As a result, the host relies on the adaptive immune response to defend against and potentially eliminate *M. abscessus*.

Line 285: I think that it should be specified/discussed here, that the Tn mutant in MAB_4691 in this study (ref. 49) was in S variant of Mab. The change in virulence in the mutant underlines the role of GL8P both in S and R variants of Mab. Moreover, the change in expression of 4690c in R and S is minimal (line 291), so the conclusion about “the importance of this compound in the pathophysiology of persistent infection by the R morphotype” (lines 292-293) would deserve stronger justification.

We apologize for the misinterpretation of the paragraph starting on line 298 (revised version). We consider that it would have been clearer to introduce a separate section specifically discussing the R morphotype (see now line 302, To our knowledge, R Δ MAB_4690c...). We added also the morphotype in front of each mutant for clarity.

While the change in expression of 4690c in both R and S variants appears minimal, this finding was observed in two independent RNAseq experiments. In light of the reviewer's comment, we have revised the sentence

to better justify the role of this compound, see lines 306-309 “Interestingly, a previous *M. abscessus*-S/R RNA-seq study³³ revealed a slight change in the expression level of *MAB_4690c* in the R compared to the S strain, further validating the importance of GL8P in the behavior of *M. abscessus* R inside the host.”, which provides a more accurate reflection of the data and its relevance to the pathophysiology of persistent infection by the R morphotype.

Line 296: “GL8Pb is built on a TAFIYLTT octapeptide with an O-methylated tyrosine”. Also Thr-8 is O-methylated (Fig. 2C, S6B, S7C). Why this is not specified here (and elsewhere?).

Upon further consideration, we agree that the methylation of Thr-8, located at the C-terminal side of the peptide, should be clearly specified. As corrected in the legends of Figures S4A and S6A, Thr-8 is predominantly carboxy-methylated, as is typical for many peptides produced by NRPS (Non-Ribosomal Peptide Synthetase) machinery in bacteria.

In contrast, we did not find examples in the literature of O-methylation occurring at the phenol group of tyrosine residues, which we found to add novelty to our study. For this reason, we propose to maintain the original phrasing in the revised manuscript to highlight this unique aspect of GL8Pb's modification.

Line 318 and 319: “GL8P represents a highly specific biomarker to consider for future diagnostic developments in *M. abscessus* infections” this statement should be supported by comparing its selectivity and specificity with other known/used Mab biomarkers.

We agree with the reviewer’s suggestion. Our primary aim in this study was to determine whether GL8P is produced *in vivo*, as evidenced by a serological response in patients infected with *M. abscessus*, which we found to be the case. Among the control patients, we identified three *M. chimaera*-positive individuals who did not exhibit a serological response to GL8P. However, this limited data does not allow us to make definitive claims about the specificity of the response. We are currently conducting additional studies using sera from patients included in the CIMeNT epidemiological study, which involves 840 CF patients, to better assess the specificity of GL8P as a potential biomarker.

To reflect this ongoing work, we have modified the sentence lines 263-266: “These results indicate that GL8Pb is recognized by sera from *M. abscessus*-infected CF patients. However, further confirmation and expansion of these results are required before considering GL8P as a reliable diagnostic marker for *M. abscessus* infections.” We also modified line 334: “potential biomarker to be considered”.

Additionally, as shown in Table S1, we observed GL8P production exclusively within the *M. abscessus* complex (*M. abscessus*, *M. chelonae*, *M. immunogenum*), which supports the specificity of GL8P production within this group. Further comparative studies with other known biomarkers for *M. abscessus* infections are necessary to establish its full diagnostic potential.

Line 387: “CHCl3 phase” – “organic phase” is more accurate

We have made the suggested modification and replaced "CHCl3 phase" with "organic phase".

Line 390-391: Please, show the profile, as I mentioned above (comments to Fig. S2)

As requested, we have added the profile and referenced it in the new Figure 3C.

Line 392-398: The details of purification procedure are missing – which were the volumes of the columns, collected fractions etc...

In response to the reviewer’s comment, we have added the relevant details of the purification procedure, including the column specifications provided by the manufacturer, the column dimensions, the gradient used, the volumes of the collected fractions, and the elution order of GL8P. We believe this additional information will assist the scientific community in replicating these experiments.

Line 411: How much of the sample was used for the analysis?

As previously mentioned, we have prioritized purity over quantity, which resulted in the recovery of relatively small amounts of the sample. However, we estimate that approximately 50 µg was sufficient to accurately determine the monosaccharide composition.

Line 427: “derivation” should be changed to “derivatization”

The term "derivation" has been changed to "derivatization" as suggested.

Line 429: Was TFA dried before addition of Marfey’s reagent? The quantities of the GL8P and the standard used for derivatization should be stated also here (not just at the Figure).

TFA was dried at 100°C prior to the addition of Marfey's reagent. Additionally, the quantities of GL8P and the standard used for derivatization have been specified in the text.

Line 780: comparison of “retention times with characterized Bacillus cereus samples” – this is not informative. Please, include more details or reference.

The phrase has been removed, as it was originally intended to reference prior work.

Reviewer #2 (Remarks to the Author):

This study identifies a novel lipopeptide from the cell wall of *M. abscessus* which the investigators refer to as GL8Pb which plays a role in the adherence of *M. abscessus* to cells and to which antibody is present in the sera from individuals with cystic fibrosis suggesting the potential for immunoassay to aid in diagnosis of *M. abscessus* infection.

Please address the following points:

1) Under section “Bacterial strains and plasmids” please list all *M. abscessus* strains used in the first paragraph, their derivation and whether they express the rough or smooth phenotype.

As requested by the reviewer, we have added a Table S2 that details all the strains used in this study. This table includes strain names, and additional details to ensure full transparency and consistency across the study.

2) The data (Figure 4A) showing Mabs CFU in THP1 macrophages need to include an initial timepoint after amikacin is removed from the culture media and the cells have been washed (time 0). The slope of growth over 72 hours appears to be identical for both wild type and the GL8Pb deletion mutant. This is in keeping with the observation in Figure 4C demonstrating significantly less adherence of the GL8Pb deletion mutant to monocytes under cold conditions which inhibit uptake. Thus, the differences observed in Fig 4A likely reflect a significant difference in initial macrophage uptake. A significant difference at time 0 comparing wild type and the GL8Pb deletion mutant would not support the conclusion of reduced intracellular survival of the GL8Pb deletion mutant in macrophages. In the zebrafish model there are significant differences between the growth of wild type and the GL8Pb deletion mutant, but this data does not allow one to determine whether these differences are due to reduced uptake by permissive cells or reduced intracellular survival.

We appreciate your thoughtful feedback and have addressed your comments with additional experiments and updates to Figure 6. We have now expanded the figure with new panels, specifically Figure 6B, 6C, 6D, 6F, and 6G, to provide a clearer representation of our findings. As requested, the initial time point post-amikacin removal and cell washing is included in panels 6A, 6B, and 6D. To demonstrate a direct impact on the mutant's growth compared to the wild-type and complemented strains, we quantified the raw integrated density at both 24 hours post-infection (hpi) and 72 hpi across 150 infected macrophages per condition. This measurement integrates both the area occupied by bacilli within each infected macrophage and the sum of red bacilli pixel values. Our analysis indicates a minor, non-significant impact on growth at 24 hpi, but a substantial delay in mutant growth at 72 hpi, supporting a significant long-term growth defect post-infection.

Additionally, we quantified the percentage of infected macrophages (Figure 6B) and assessed attachment efficiency (Figure 6F), which indicate a clear adhesion defect in the mutant, reflected by the significant drop in intracellular bacterial load at 4 hpi (shown in Figures 6A and 6D). Taken together, our findings demonstrate that GL8P is important for both macrophage adhesion and long-term growth within infected macrophages.

In the zebrafish model, the reduced bacterial load observed at 4 days post-infection (dpi), which further supports a role for this lipid in bacterial survival and/or optimal growth *in vivo*.

Please make clear in Figure 5A and 5B how many mice are in each group. Although mouse numbers are mentioned, it is hard to correlate what is in the Figure legend with the data in the Figures. Similarly, make clear whether the data showing organ CFU are in C3HeB/FeJ or BALB/c mice. Please explain in results or discussion why the survival of wild type mice is significantly reduced at approximately day 18 (Figures 4A,B) compared to the GL8Pb deletion mutant, yet the difference in organ CFU comparing wild type to percentage of mice surviving is not significantly different until Day 30 (Figs C-F). Also, please provide an explanation as to why growth seems to only increase in the kidneys of mice whereas in the liver, spleen and lungs it remains stable and then declines in all bacterial groups near the end of the experiment. Considering these observations, please comment as to the cause of death in infected mice.

We apologize for any confusion caused by the figure presentation. To clarify, the number of mice in each group is indicated twice: $n = 11$ per group for both Figure 7A and 7B. Every time a mouse dies, the curve decreases by approximately 9%. Additionally, all CFU counts were performed using C3HeB/FeJ mice, as indicated in the figure legend, while data obtained in BALB/c mice are provided in the Supplementary Figure 7. The doses used in each experiment are indicated in the figure legends: 10^8 for survival experiments and 10^6 for CFU determination.

Although the primary aim of this manuscript was not to explore the detailed pathophysiology of *M. abscessus* infection in mice, we acknowledge the reviewer's point. Previous studies (doi: 10.1093/infdis/jit614; doi: 10.1128/IAI.00835-06) have reported similar phenomena, suggesting that the kidney serves as a reliable marker for assessing the therapeutic efficacy of anti-*M. abscessus* antibiotics. Furthermore, in Catherinot *et al.*'s work (doi: 10.1128/IAI.00835-06) (ref 21 in the text), renal damage was linked to disseminated intravascular coagulation, leading to the death of mice through multivisceral failure.

Please comment and/or note that a limitation in terms of drawing conclusions regarding Mabs mouse infection with the intravenous infection model is that this utilizes a very high inoculum (106) and does not mimic human lung infection. There are now other mouse models of lung infection, which are more clinically relevant to human infection (Tuberculosis, Volume 147, July 2024, 102503).

We fully agree with Reviewer 2 that alternative mouse models of infection exist, and we have been actively involved in the development and use of several of these models in the past. For example, in studies by Le Moigne *et al.* (doi: 10.1016/j.vaccine.2015.03.030; doi: 10.1080/21645515.2015.1102810; doi: 10.1128/IAI.00359-16), we developed a respiratory infection model, and in studies involving Bedaquiline and Imipenem treatments, we employed the agar bead model (Le Moigne *et al.*, doi: 10.1128/AAC.00114-20). Each of these models serves to address specific research questions.

The primary aim of this work was to establish an immunocompetent and stringent model that would enable to differentiate an avirulent mutant from its parental strain. The "Kramnick" C3He/FeJ model, using appropriate infectious doses, provided this possibility. Moreover, the doses used here are fully consistent with those described in the reviewer's referenced article.

Please indicate whether it is known whether rough or smooth (or mixed isolates) were obtained from sputum cultures of patients with CF who had antibody to GL8Pb detected in their serum.

We have conducted an epidemiological study in France on the prevalence of NTM infections in cystic fibrosis (CF) patients, and we have an associated biocollection that includes sputum samples. As we compile the data for each patient, we will be able to address this specific question regarding the presence of rough or smooth (or mixed) isolates in sputum cultures from patients who have seric antibodies against GL8Pb.

However, it is important to note that, in Europe, liquid culture media are commonly used for inoculating sputum after decontamination, which may not facilitate the differentiation of morphotypes typically observed on solid media. However, it is very likely that patients harbor both rough and smooth variants, as suggested in previous epidemiological studies cited earlier.

The discussion related to antibody to Mabs exerting selective pressure on Mabs in terms of smooth and rough phenotypes needs to be supported by data. Please supply references indicating that antibody to Mabs has been shown to result in bacterial killing, for example by complement fixation. If no data is available, please modify discussion accordingly. It is established that antibody may have negative or positive effects on pathogen survival (Infection and Immunity April 2021 Volume 89 Issue 4 e00054-21).

We thank the reviewer for their insightful comment and for suggesting the reference. To clarify this point, we are not discussing bacterial killing *per se*, but rather the selective pressure exerted by antibodies produced by CF patients during *M. abscessus* infection over extended periods of time. This pressure may drive the emergence of different bacterial morphotypes.

C-type GPLs have been well-documented for their immune-modulatory properties, eliciting strong antibody responses (Chatterjee and Khoo, 2001). The significance of GPLs in the host-pathogen interaction of *M. abscessus* is further underscored by the persistence of the ATCC19977 R variant in μ MT-KO mice, which lack an efficient B-cell response (Rottman et al., 2007) (ref 45 line 281). While immune-competent mice are able to clear intravenous infections with smooth *M. abscessus*, μ MT-KO mice are unable to efficiently eliminate these bacteria, which increases the likelihood of genetic changes in the smooth variants of *M. abscessus*, potentially leading to the emergence of rough variants.

We have now incorporated the reference to Rottman et al. at the beginning of line 281 to provide additional context and support for the role of the immune response in driving phenotypic variation in *M. abscessus*.

In light of the above considerations, the title - "An exclusive glycosylated lipooctapeptide governs rough Mycobacterium abscessus virulence" is confusing and overstates the findings. The use of the word exclusive does not make sense in the context of the findings. What is being excluded? Do the authors mean novel or unique? In addition, the word governs does not describe the relationship of the lipooctapeptide to M. abscessus virulence. Governing implies overarching control. Other bacterial components/products are also involved in Mabs virulence and the data supports a role for GL8Pb in cellular uptake.

Thank you for your insightful feedback regarding the title. We appreciate your concerns about the terminology used and agree that "exclusive" may not accurately represent our findings. Additionally, we recognize that the word "governs" could imply an overly dominant role for the glycosylated lipooctapeptide in *M. abscessus* virulence. To clarify our findings and better reflect the role of GL8Pb in cellular uptake and growth, we propose the following revised title: **"A unique glycosylated lipooctapeptide promotes uptake and sustained growth of rough *Mycobacterium abscessus* in the host"**. We believe this new title more accurately describes our work while addressing the reviewer's concern.

Reviewer #3 (Remarks to the Author):

Mycobacterium abscessus is a medically important pathogen with established lipoglycan-dependent phenotypes. Here, Leclercq and coauthors identify two genes of unproven function, demonstrate that they encode factors necessary and sufficient for the biosynthesis of previously unknown glycopeptidolipids (GPLs), determine the GPL structures in detail, and demonstrate clear and complemented virulence effects in cellular and in vivo models. Further, a small study demonstrates GPL-specific human immunoglobulin responses, supporting possible species specificity of the glycans identified. The experiments are generally generated with high rigor with complementation of key phenotypes. Despite overall high enthusiasm for the originality and breadth of work, we have major concerns about certain interpretations and data display. The paper could be improved with a less-chemistry focused data presentation to make the paper comprehensible to physician, bacteriologist, and immunologist readers of Nature Communications.

Major

Some discussion of glycan biosynthesis and smooth/rough (S/R) phenotypes as motivating data is reasonable. However, lengthy discussions both before (p. 3) and after the data presentation (p. 12) pose several problems. The sections are redundant, and they set unfulfilled expectations, as the data do not address "intense remodeling" and/or morphotypic change. The discussion of TLR function (line 261) is also a non sequitur. The authors should either discard much of this detail (especially the discussion on p.

12), or include additional data answer the questions raised: what is the yield of these GPLs relative to other GPLs and/or relative to total cell mass? Do deletions in 4690c and 4691c lead to altered morphology?

As requested by the reviewer, we have deleted “intense remodeling” from the text. Additionally, we have removed some details from the discussion in the revised version.

The cellular and animal model conclusions must be stated more carefully. Not all zebrafish phenotypes are macrophage-dependent, so the unqualified attribution of outcomes to macrophages (line 218) is invalid.

Thank you for your valuable feedback. We recognize that not all zebrafish phenotypes are exclusively dependent on macrophages, and we agree that attributing all outcomes solely to macrophage activity in line 218 (first version) was overly broad. Consequently, we have revised the text by removing the latter part of sentence 218 (now lines 237 to 238 in the revised version).

THP-1 cells are immortalized monocytic cells (not macrophages). Also, the use of these models is inadequately justified; additional references and discussion are needed.

Our choice to use THP-1 cells is grounded in their proven effectiveness for studying the uptake and survival of mycobacterial mutants, as well as transgenic strains that overproduce specific proteins of *M. abscessus*. For example, in Roux *et al.* (*Open Biol.*, 2016), we demonstrated the distinct intracellular fates of smooth and rough *M. abscessus* variants within THP-1 cells, showing that these cells are suitable for dissecting macrophage-specific responses to *M. abscessus* phenotypes. Additionally, in Daher *et al.*, *CCB*, 2022, we conducted parallel experiments using both human primary macrophages and THP-1 cells to investigate the role of glycopeptidolipid glycosylation in *M. abscessus* pathogenicity, with very similar results across both cell types. We have now included additional references to support our model choice in the Methods section.

How do these models relate to human disease? If they were chosen based on S/R virulence phenotypes, is there a clear relationship between the S/R morphotype switch and clinical outcomes?

Thank you for this insightful comment regarding the relevance of our *in vivo* models to human disease, particularly with respect to the *M. abscessus* Smooth (S) and Rough (R) morphotypes. Both zebrafish and mouse models were carefully chosen for their ability to recapitulate essential aspects of *M. abscessus* infection dynamics and pathogenesis, making them well-suited for exploring the implications of the S/R morphotype switch in a way that aligns with clinical outcomes.

The zebrafish model is highly suitable for studying early host-pathogen interactions, especially granuloma formation, which is a hallmark of mycobacterial disease in humans. Previous research demonstrated that the R morphotype is more invasive and causes severe granulomatous lesions, reflecting the aggressive and chronic progression often observed in human patients with the R form. Thus, the zebrafish model effectively mirrors the transition from colonization to invasive disease that is clinically significant.

Complementing this, the mouse model allows us to examine the later stages of infection and immune responses, including the development of granulomas and adaptive immunity. Studies from our labs have shown that mice infected with the R morphotype exhibit enhanced tissue invasiveness, immune evasion, and prolonged infection. By focusing on the R morphotype in both zebrafish and mouse models, we are able to study the most pathogenic form of *M. abscessus*, which is directly relevant to disease progression in humans.

Please see also references: doi: 10.1073/pnas.1321390111 ; doi: 10.1128/AAC.00142-14 ; doi: 10.3791/53130 ; doi: 10.1128/IAI.00359-16 ; doi: 10.1371/journal.ppat.1005986 ; doi: 10.3389/fcimb.2017.00100 ; doi: 10.1073/pnas.1812984115 ; doi: 10.1371/journal.ppat.1010771 ; doi: 10.1128/AAC.00114-20 ; doi: 10.1073/pnas.1713195115 ; doi: 10.1073/pnas.1605477113.

The main figures should show at least some data for every major claim. However, key data showing the effect of gene deletion on glycopeptide production is in Figure S2. In addition, the title states the GPLs are “exclusive” (which should possibly be rephrased as “species specific”): the authors should consider presenting species analyses (Figure S1) as a main figure. Color coding in this figure must also be added.

Thank you for your constructive comment, which was also raised by the other reviewers. In response to this comment, we have updated the title of the manuscript to more accurately reflect the species-specific nature of the glycolipid production, as suggested. Additionally, we have moved the key data showing the effect of gene deletion on glycolipid production from Figure S2 to the main text (now Figure 3), ensuring that all major claims are supported by data in the primary figures. Regarding Figure S1, we agree that it is an important information and have now included it as a main figure (Figure 2), with color codes added for clarity, as suggested by the reviewer.

Conversely, detailed multidimensional NMR data shown in main (Fig. 2) will not be comprehensible to most generalist readers and could move to supplemental.

We agree that some of the details of the interpretation of the NMR data may not be comprehensible to all readers, but we believe that solid and unambiguous data are absolutely essential to demonstrate the validity of our structural hypothesis, as the compound is completely new. In response to the reviewer's suggestion, we have chosen to retain only the most informative NMR spectra in a single figure out of the height presented. Specifically, Figure 4A highlights key data on amino acid composition and peptide sequencing, while Figure 4E illustrates the position of glycosidic bonds. All additional NMR data, which are essential for the *de novo* structural identification, but may be more technical for generalist readers, have been moved to the Supplementary materials and tables to ensure that they remain accessible to those interested in the full dataset.

Additionally, certain chemical interpretations that are apparent to the authors will be inaccessible to most readers. Why do epimerases map specifically to the 1, 3, 6, and 8 positions? What are Marfey derivatives?

As mentioned above, the data cited are also essential and will be appreciated by those readers who are experts in the analysis of bioactive molecules, without hindering the overall understanding of this multidisciplinary study. To specifically respond to the reviewer's comment, the mapping of epimerases to the 1, 3, 6, and 8 positions is based on predictions from the Antismash analysis of the NRPS modules of *MAB_4690c* and *MAB_4691c*. Part of the study's objective was to validate these *in silico* predictions through structural data. Regarding Marfey's derivatives, they are a well-established method for distinguishing between L- and D-amino acids. The derivatization with L-fluoro dinitro phenyl alaninamide (L-FDAA) generates distinct derivatives that can be separated by reverse-phase HPLC, as demonstrated in Figure 4B and supported by literature (e.g., *J. of Chromatography* 10.1007/s00726-004-0118-0 and 10.1007/s13361-018-2093-9). We have clarified these points in the legend of Fig. 4 in the revised manuscript to improve accessibility for readers.

Why does deshielding of H-2 carbon at 80.77 suggest alpha-rhamnopyranose?

As stated in the original manuscript, the deshielding of the H-2 carbon at 80.77 ppm suggests the presence of 2-O-methyl- α -rhamnopyranose. This is due to the fact that methylation of hydroxyl groups on monosaccharides alters their ^{13}C chemical shifts, as detailed in the review by Li *et al.* (10.1016/j.carbpol.2021.118885), which discusses the impact of various modifications on NMR spectra. In our study of GL8Pb, we identified two rhamnosides residues using GC-MS, and NMR analysis was crucial for determining their specific order. These are also important and appreciated elements for the readership.

What are a2 to a6 fragments?

The "a2 to a6 fragments" refer to ions observed in mass spectrometry (MS) fragmentation, specifically along the peptide backbone. This nomenclature is widely used in proteomics for protein sequencing and identification. The "a" fragments represent ions derived from the N-terminal side of the peptide, formed by cleavage between the α -carbon and carbonyl groups. In contrast, "y" fragments arise from the C-terminal side, resulting from fragmentation between the carbonyl and amide groups. For further clarification, this is explained in the Nature Biotechnology review (10.1038/s41587-022-01424-w).

What are M-146 and M-164 fragment ions?

We apologize for not specifying the atomic mass units (a.m.u.) for the fragment ions. The M-146 and M-164 ions correspond to the loss of a terminal rhamnose residue during the fragmentation of GL8Pa around the glycosidic bond. Specifically, the M-146 ion results from the loss of a putative terminal rhamnose residue, as indicated by the decrease of 146 a.m.u. in the mass spectrum. We have clarified this point in the revised manuscript.

Authors should remove or define undefined acronyms: COSY, NOESY, HSQC, HMBC, TOCSY, NOE, CFU, hpi, GDBP, C/M/W and many others.

We agree that clarity is essential, and we have carefully addressed this concern by defining and explaining all acronyms throughout the Material and Methods section of the revised manuscript. The full definitions are as follows:

- MALDI QIT-TOF (Matrix Assisted Laser Desorption and Ionization with Quadrupole Ion Trap and Time Of Flight)
- FT-ICR (Fourier Transform Ion Cyclotron Resonance)
- GC-MS (Gas Chromatography coupled to Mass Spectrometry)
- LC-MS (Liquid Chromatography coupled to Mass Spectrometry)
- a.m.u. (atomic mass unit)
- COSY (COrrrelation SpectroscopY)
- NOESY (Nuclear Overhauser Effect SpectroscopY)
- HSQC (Heteronuclear Single Quantum Coherence)
- HMBC (Heteronuclear Multiple Bond Correlation)
- TOCSY (TOtal Correlation Spectroscopy)
- CFU (Colony-Forming Units)
- hpi (hours post-infection)
- C/M/W (CHCl₃/CH₃OH/H₂O)

These terms are now fully defined in the manuscript to ensure clarity for readers.

The cooling assay in Fig. 4C is unusual. It is not explained in the results section and is lacking cites to support its use as an adhesion assay. Please consider repeated using cytochalasin D or another literature-supported technique to block phagocytosis (see Toniolo et al. 2023, EMBO J. 42:e113490).

As suggested by the reviewer, we have performed additional experiments and updated the figure with new panels (now Figures 6F and 6G). We have also expanded the *Materials and Methods* section to provide detailed descriptions of our attachment assays, incorporating cytochalasin D experiments. Additionally, we have cited relevant literature within the manuscript, including the recommended Toniolo et al. (2023) reference, as well as an additional reference (Dubois et al., PNAS, 2018). Our results demonstrate that cytochalasin D has no significant impact on bacterial attachment (Figure 6F), while it drastically reduces internalization (Figure 6G), confirming the specificity of the attachment assay and the results observed.

Why wasn't the genetic complement used in Balb/c mice?

We apologize for the oversight in the previous Figure 5B, where the survival data for both the $\Delta 4690c$ and $\Delta 4691c$ mutants were not clearly presented. This has been corrected in the revised manuscript.

Regarding the use of genetic complementation in Balb/c mice, we focused on the $\Delta 4690c$ mutant, for which complementation was performed and CFU counts were assessed (Fig. S7). Since the phenotype of the $\Delta 4691c$ mutant mirrored that of the $\Delta 4690c$ mutant in terms of survival and GL8P production (Fig. S7A), and considering the extremely large gene size of the $4691c$ gene (>24 kb, preventing its cloning in a complementation plasmid), we did not attempt to complement mutant. As a consequence, we focused essentially on the $\Delta 4690c$ mutant, which provided sufficient data to address our study's objectives.

MINOR

There is very limited discussion of the human immunoglobulin experiment (Figure 6), such as next steps. Also, ELISAs have been highly non-specific for diagnosis overall, yet there is fairly good specificity here. Does this have to do with surface localization or glycans as particularly good immunoglobulin targets?

As highlighted in the manuscript and in response to Reviewer 2, the diagnostic potential of the human immunoglobulin experiment (now Figure 8) is still under active investigation. Specifically, we are compiling data from a cohort of 840 French cystic fibrosis (CF) patients as part of a survey on the prevalence of non-tuberculous mycobacterial (NTM) infections in this population.

We have removed the term "specificity," as our primary aim was not to establish a diagnostic test but to investigate whether GL8P is produced during *M. abscessus* infection in CF patients, as evidenced by the levels of antibodies targeting GL8P. The results observed in the control group of CF patients without *M. abscessus* infection (with a subset infected with *M. chimaera*, which did not test positive for GL8P antibodies) further support this interpretation. We have amended the manuscript accordingly to clarify this distinction.

Additionally, while the ELISA data show promising results, we believe that the observed antibody recognition may be linked to the surface localization of GL8P (or its glycan moiety), which could make it an effective target for the immune response.

Various statistical tests are described in methods without explaining which were used for each figure. Figure-specific statistical test descriptions are needed for Nature style, but are missing for Figure 5.

We have now included the specific statistical tests used for each figure in the corresponding figure legends, as well as in the Materials and Methods section. For new Figure 7, the appropriate statistical methods are as follows:

- Survival data (Panels A and B): P values were determined by the Log-rank (Mantel-Cox) test for Kaplan-Meier survival curves to assess statistical significance.
- Other data (Panels C to F): P values were calculated using an unpaired t-test, as indicated in the figure legend. * $P < 0.05$, ** $P < 0.01$.

We have also clarified these points in the Materials and Methods section for consistency.

In Fig. 4, panels need to appear alphabetical order. Significance indicators in 4D need to be shifted.

We have made substantial updates to new Figure 6, adding several new results while ensuring that all graphs now appear in alphabetical order. Additionally, we have adjusted the significance indicators, which are now correctly positioned in Figure 6H (formerly Figure 6D). We appreciate the reviewer's input, which has helped us improve the clarity and presentation of the data.

Fig. 4B microscopy needs quantitation of organisms. Similarly, Fig. 4A CFU data are used to support a claim of impaired growth at later timepoints; this would require an additional plot showing ratios of CFU counts at 24 and 72 hours vs. 4 hours.

As mentioned above we have expanded figure 6 to include additional panels, specifically Figures 6B, 6C, 6D, 6F, and 6G, which now provide a clearer representation of our findings. In Figures 6B and 6C, we have included quantification of the organisms observed in the microscopy images, addressing your request for a more detailed analysis. Our findings indicate a minor, non-significant impact on growth at 24 hours post-infection (hpi); however, there is a substantial delay in mutant growth observed at 72 hpi, supporting our claim of a significant long-term growth defect post-infection. Together, these updates enhance the clarity and robustness of our results, emphasizing the role of GL8P in macrophage adhesion and optimal long-term growth within infected macrophages.

The claims related to Fig. 5C-F could be softened, since the observed CFU differences are mild and intermittent.

We agree that the observed CFU differences are mild and intermittent, but we believe that the reduced killing activity in mice with these mutants is a significant finding. As this is only the second R mutant showing strong attenuation in mice, we felt this warranted some emphasis. In response, we have softened the claims by removing the word "highly" from the title of the paragraph and revising line 238 to avoid overstating the impact.

OD values in ELISA usually show loss of sensitivity after OD values of 2, yet many values that distinguish the groups are in the 3-4 range. Are these correct? Could better discrimination be obtained by diluting samples?

We appreciate your observation regarding the OD values, and we acknowledge that ELISA assays often show reduced sensitivity at OD values above 2. As can be seen in Figure 8, the values distinguishing the groups fall within the 3-4 OD range, which we have observed in similar studies (Le Moigne et al., doi: 10.1128/spectrum.00192-22; doi: 10.1016/j.jcf.2021.08.019). We will further investigate this within our ongoing cohort of French CF patients to determine whether it reflects two distinct patient populations with varying disease severity. Additionally, we would like to highlight that the limited availability of GL8P for testing may have contributed to this outcome. We are currently awaiting the synthesis of GL8P through an international collaborative project, which should enable us to refine the assay and potentially improve discriminations.

The strain designations change from figure to figure, and the single letter designations used in the supplemental figures are not sufficiently explicative for the reader.

We acknowledge that consistency in labeling is crucial for clarity and comprehension. To address this issue, we have revised all figures to ensure that strain designations remain consistent throughout the manuscript. Additionally, we have clarified the single-letter designations used in the supplemental figures by providing a legend that explains each designation in detail. We hope this will help the reader to better follow the content of the study and strains used all throughout the manuscript.

Reviewer #1 (Remarks to the Author):

The authors addressed all of the raised issues to my satisfaction. Nevertheless, I still propose a few minor modifications for their consideration.

(i) Although the names tri-O-acylated phosphatidyl-inositol dimannoside for Ac1PIM2, and tetra-O-acylated phosphatidyl-inositol dimannoside for Ac2PIM2 are still widely used in the literature, I consider the name (mono)acylated phosphatidyl-inositol dimannoside for Ac1PIM2 as proposed by Kordulakova et al., 2003 (DOI: 10.1074/jbc.M303639200), and, consequently, di-acylated phosphatidyl-inositol dimannoside for Ac2PIM2, as more accurate. (The name phosphatidyl-inositol is given to the molecule, which already has 2 acyl groups, so tri-O-acylated phosphatidyl-inositol dimannoside would contain 5 acyls, which is not the case.)

The reviewer is absolutely right that the numbers (AcxPIMy) differ from the actual number of acyl groups, which may cause confusion. We have applied the suggested nomenclature in the main text and the legend of Fig.3.

(ii) I appreciate correction in labelling the lipids in Fig. 3B, but I did not see the proof of identification of CL, although it is mentioned in the author's reply. I am surprised that other common phospholipids, such as phosphatidyl ethanolamine or phosphatidyl glycerol, which I would expect to migrate between DAT and PIMs (eg. according to Chiaradia et al., 2017, DOI: 10.1038/s41598-017-12718-4; Fig. 7) were not detected in the polar lipid fraction.

Cardiolipins and other common phospholipids were indeed detected during the course of the study. However, as these compounds were not relevant to the present study and no major differences were observed between the strains, we decided not to detail their structures to maintain conciseness in an already dense manuscript, in line with another reviewer's comments. Nevertheless, we have added this information to the main text and the legend of Figure 3.

In the Figure S1, in which characterization of PIM species is provided, the information in the legend is lacking mentioning the R3 substituent. I think that also in the legend to Figure S1, it would be helpful to mention that Man and methylated Man are present in PIM2 species.

The nature of the R3 substituent and the monosaccharides have been added to the legend of Figure S1, as requested.

(iii) In the new Figure S5C (upper part showing the fragmentation pattern of the molecule), the authors replaced number 129 by 170. My concern was that in the original fragmentation pattern, 170 was missing, yet it was present in the spectrum. Now the situation is opposite – number 170 is in the fragmentation pattern, but 129 is missing there, but present in the spectrum.

In the previous version, the attribution of the ion at m/z 129 was incorrect, as the reviewer rightly pointed out. This has been corrected in the present version by replacing it with the ion at m/z 170. The ion at m/z 129 is a secondary fragment that may have multiple origins and thus does not provide essential information. Although it is observed in the spectrum, we

believe it is better not to interpret it on the fragmentation pattern to avoid potential misassignment.

Reviewer #2 (Remarks to the Author):

The investigators have adequately addressed Reviewers concerns.

Reviewer #3 (Remarks to the Author):

Comments for the Authors

Leclercq et al. have meaningfully improved upon their technically sound manuscript by reworking figures, adding the requested controls for THP-1 experiments, and addressing some interpretive issues. However, three major interpretative issues remain:

1. The authors chose infection models based on S/R phenotypes, with the key premise that the R form is the more virulent type of *M. abscessus* in humans. Only 2 cited multi-patient studies (Jönsson et al. 2007 & Hedin et al. 2023) posit key conclusions about R vs. S virulence, and these are based on low patient numbers, so we are not convinced that there is a clear relationship between the S/R morphotype switch and clinical outcomes. Unless the authors can cite unambiguous clinical data supporting the increased human-tropic virulence of the R forms of *M. abscessus*, the infection models do not directly inform human "virulence" in natural disease but are nevertheless useful in vivo phenotypes in defined experimental settings in cells and zebrafish. We therefore ask that the authors avoid reference to GL8P as a "virulence lipid" in the title and throughout the manuscript.

Morphotype and virulence are connected features that apply to several pathogenic mycobacterial species. For instance, rough strains of *Mycobacterium canetti*, result from genetic events causing the loss of lipooligosaccharide biosynthesis and subsequent morphotype changes. These rough variants showed an altered host-pathogen interaction and increased virulence in cellular- and animal-infection models. *M. tuberculosis* isolates with a well-known rough morphotype present a similar genetic event, and well-known as *M. tuberculosis* lack lipooligosaccharide. Both belong to the *Mycobacterium tuberculosis* complex (Boritsch et al., Nat Microbiol, 2016 cited in the manuscript). The same applies to *M. kansasii*, which appears rough when cell surface lipopolysaccharides are absent and persists longer than smooth variants in experimentally infected mice (PMID: 2722755). The morphotypes of the *M. avium* complex, the *M. abscessus* complex, and rough isolates of *M. chelonae* are other examples of this physiological adaptation, which has increased the virulence of these mycobacteria in cell lines, animals, and humans. In a recent multicenter study, Hedin *et al.* showed that among 71 *M. abscessus* isolates from patients in Sweden (2009-2020), 23 were rough. Patients with rough colony morphology isolates had worse clinical outcomes compared to those with smooth isolates (PMID: 36637124).

Most, clinicians treating *M. abscessus* patients are aware that the emergence of the rough (R) morphotype during the course of the disease is linked to increased severity in humans (See Byrd and Lyons *et al.* 1999; and Howard et al., Microbiology, 2006). It is also important to keep in mind that the global number of *M. abscessus* patients worldwide is much lower than that of patients with tuberculosis, resulting in very few epidemiological surveys to date.

In addition, there are two difficulties associated with epidemiological studies. The first is that the S-to-R conversion is irreversible and mostly occurs during the course of human disease.

Currently, there is no evidence of direct infection with the R morphotype, most probably because the S morphotype represents the environmental and colonizing form. However, isolation of the S variant from the environment is rare, and no studies have documented the isolation of the R morphotype in the environment.

The second difficulty is diagnostic. The media used to diagnose NTM infections are liquid media, in which both morphotypes can co-exist, with the S morphotype multiplying more rapidly than the R morphotype. This complicates studies and makes interpretation of results more challenging.

Reference 6 in our manuscript illustrates disease exacerbation linked to morphotype transition, with strictly R morphotype colonies being isolated from patient samples when disease exacerbated.

Additionally, several cellular and animal models developed by us and others have confirmed the hyper-virulence of the R morphotype compared with the smooth morphotype (refs 20, 21 for example cited in the manuscript, or Kam et al., Nat. Comm.2022). In most of these studies, the R morphotype resulted from the loss of surface glycolipids. Thus, for all these reasons, we believe that GL8P contributes to the virulence of *M. abscessus*.

2. This paper has the potential for interdisciplinary appeal to microbiologists, infectious disease physicians, immunologists, and chemists. Yet, the revised version will likely lose readers due to much too much subspecialty jargon and interpretation that occurs without any explanation to the non-chemist reader, especially in the abstract and lines 132-201. The question even arises as to whether such subspecialty-focused writing will have broad appeal required by Nature Communications editors.

Since we present the discovery of a new glycolipid, extensive biochemical and structural studies were conducted to determine the detailed structure of GL8P and to assess how it differs from well-known glycopeptidolipids (GPL). We do not believe this will affect the broad appeal required by Nature Communications, as much of the data presented in this manuscript also focuses on the function of GL8P in host-pathogen interactions and its role in infected mice. Moreover, we propose translational applications of GL8P as a potential diagnostic marker. Overall, this manuscript provides a complete and compelling story that will be of high interest to the broader mycobacterial research community.

Tangible suggestions: 1) provide a clear definition for every symbol in the acronym GL8P (abstract) and remove the confusing acronym BGC.

As suggested by the reviewer, we have defined GL8P, which stands for glycosylated lipooctapeptides, in the abstract.

We have removed the confusing acronym BGC.

Previously, we asked questions like "what are Marfey derivatives?" and "what are a2 to a6 fragments?" to elicit changes to lines 132-201. Yet the authors replied to us directly rather than amending the paper on the readers' behalf - answers to these several questions asked in prior review should now be provided in a further revised manuscript.

As suggested by the reviewer, further explanations and references to the methodologies used have been included in the main text, complementing those added in the previous version, to enhance accessibility for non-specialist readers.

3. There is widespread agreement regarding major differences between primary tissue 'macrophages' and the 'transformed monocytic cell line, THP-1.' The best way to respond to prior criticisms of conflating these two very different terms would be to show that GPL knockouts also have altered survival in primary macrophages. Lacking this information, the authors might consider a 'limitation of the study' statement regarding the lack of primary macrophage data. The many instances in which THP-1 cells are designated as macrophages (abstract, line 210, 221, 222, other) are substantively inaccurate and should be amended to say "THP-1 cells."

We appreciate your insightful comments and the opportunity to clarify our use of THP-1 cells in this study. We acknowledge the significant differences between primary tissue macrophages and the transformed monocytic cell line, THP-1, and recognize the importance of avoiding conflation of these distinct cell types.

In response to your suggestion, we have carefully revised the manuscript to ensure that all references to THP-1 cells are accurately described as such. Specifically, we have amended the abstract, as well as lines 210, 221, 222, and other instances throughout the manuscript, replacing the term "macrophages" with "THP-1 cells."

Regarding the inclusion of primary macrophage data, we fully agree that such experiments would provide valuable additional evidence to strengthen our findings. While we currently lack survival data for GL8P knockouts in primary macrophages due to the scope and resources of this study, we have added a clear statement in the revised manuscript, under the "Discussion" section, acknowledging this as a limitation. Additionally, we have emphasized the importance of future studies to investigate whether the observed phenotypes in THP-1 cells extend to primary macrophages.

Minor

The description in the Methods of the new "raw integrated density" measure does not specify that it focuses just on infected host cells (as the authors explained to Reviewer 2 in the response letter). They should add this detail so readers can understand exactly what is being compared in Fig. 6C.

We have revised the Methods section to explicitly state that the "raw integrated density" measure specifically focuses on infected host cells, as previously explained in our response to Reviewer 2. This additional detail will ensure that readers have a clear understanding of what is being compared in Figure 6C and the context in which this measure is applied.